# SG-I2V: Self-Guided Trajectory Control in Image-to-Video Generation

**Koichi Namekata**[1], **Sherwin Bahmani**[1,2], **Ziyi Wu**[1,2], **Yash Kant**[1,2], **Igor Gilitschenski**[1,2],
**David B. Lindell**[1,2]
[1]University of Toronto, [2]Vector Institute

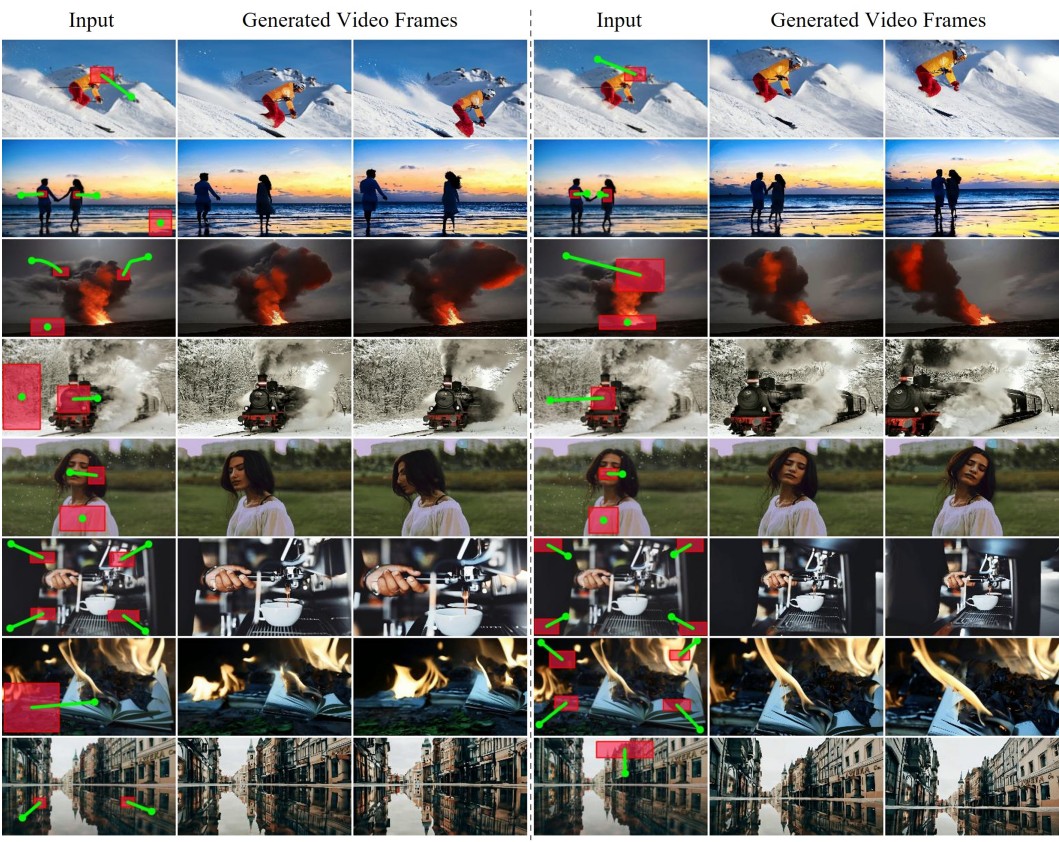

Figure 1: **Image-to-video generation based on self-guided trajectory control.** Given a set of bounding boxes with associated trajectories, we achieve object and camera motion control in image-to-video generation by leveraging the knowledge present in a pre-trained image-to-video diffusion model. Our method is self-guided, offering zero-shot trajectory control without fine-tuning or relying on external knowledge.

## ABSTRACT

Methods for image-to-video generation have achieved impressive, photo-realistic quality. However, adjusting specific elements in generated videos, such as object motion or camera movement, is often a tedious process of trial and error, e.g., involving re-generating videos with different random seeds. Recent techniques address this issue by fine-tuning a pre-trained model to follow conditioning signals, such as bounding boxes or point trajectories. Yet, this fine-tuning procedure can be computationally expensive, and it requires datasets with annotated object motion, which can be difficult to procure. In this work, we introduce SG-I2V, a framework for controllable image-to-video generation that is self-guided—offering zero-shot control by relying solely on the knowledge present in a pre-trained image-to-video diffusion model without the need for fine-tuning or external knowledge. Our zero-shot method outperforms unsupervised baselines while significantly narrowing

down the performance gap with supervised models in terms of visual quality and motion fidelity. Additional details and video results are available on our project page: `https://kmcode1.github.io/Projects/SG-I2V`.

# 1 INTRODUCTION

Recent advances in video diffusion models demonstrate significant improvements in visual and motion quality (Ho et al., 2022b; Blattmann et al., 2023a;b; He et al., 2022). These models typically take a text prompt (Ho et al., 2022b; Blattmann et al., 2023a; Ho et al., 2022a) or image (Chen et al., 2023; 2024a; Guo et al., 2024; Xing et al., 2024) as input and generate video frames of a photorealistic, animated scene. Current methods can generate videos that are largely consistent with an input text description or image; however, fine-grained adjustment of specific video elements (e.g., object motion or camera movement) is conventionally a tedious process that requires re-running the model with different text prompts or random seeds (Wu et al., 2024b; Qiu et al., 2024).

Approaches for controllable video generation aim to eliminate this process of trial-and-error through direct manipulation of generated video elements, such as object motion (Wu et al., 2024c; Yin et al., 2023; Wang et al., 2024a), pose (Hu, 2024; Xu et al., 2024b), and camera movement (Wang et al., 2024c; Li et al., 2024; He et al., 2024a; Hu et al., 2024). One line of work fine-tunes pre-trained video generators to incorporate control signals such as bounding boxes or point trajectories (Wu et al., 2024c; Wang et al., 2024c). One of the primary challenges with these supervised methods is the expensive training cost, and thus, previous methods usually incorporate trajectory control by fine-tuning at a lower resolution than the original model Wu et al. (2024c); Yin et al. (2023). More recently, several methods for zero-shot, controllable text-to-video generation have been developed (Ma et al., 2023; Qiu et al., 2024; Jain et al., 2024). They control object trajectories by modulating the cross-attention maps between features within a bounding box and an object-related text token. Still, it is not always possible to associate a desired edit with the input text prompt (consider, e.g., motion of object parts). Moreover, these methods cannot be directly applied to animate existing images, as they are only conditioned on text.

In this work, we propose SG-I2V, a new method for controllable image-to-video generation. Our approach is *self-guided*, in that it offers zero-shot control by relying solely on knowledge present in a pre-trained video diffusion model. Concretely, given an input image, a user specifies a set of bounding boxes and associated trajectories. Then, our framework alters the generation process to control the motion of target scene elements. It is essential to manipulate the structure of the generated video to achieve precise control over element positions, which is mainly decided by early denoising steps (Balaji et al., 2022; Wang & Vastola, 2023). In image diffusion models, it is known that feature maps extracted from the output of upsampling blocks are *semantically aligned*, i.e., pixels belonging to the same object share similar feature vectors on the feature map and thus can be used to control the spatial layout of generated images (Tang et al., 2023; Shi et al., 2024; Namekata et al., 2024; Tumanyan et al., 2023). However, our analysis reveals that feature maps extracted from the upsampling blocks of video diffusion models are only weakly aligned across frames (see Fig. 2). This misalignment poses challenges, as directly manipulating these feature maps fails to give useful guidance signals for layout control. Instead, we find that feature maps extracted from the self-attention layers can be semantically aligned by replacing the key and value tokens for each frame with those of the first frame (see bottom row of Fig. 2). After that, we can control the motion of generated videos by optimizing the latent (the input to the denoising network) with a loss that encourages similarity between the aligned features within each bounding box along the input trajectory. Finally, we apply a post-processing step to enhance output quality by ensuring that our optimization does not disrupt the distribution of high-frequency noise expected by the diffusion model.

In summary, our work makes the following contributions:

- We conduct a first-of-its-kind analysis of semantic feature alignment in a pre-trained image-to-video diffusion model and identify important differences from image diffusion models.

- Building on this analysis, we propose SG-I2V, a zero-shot, self-guided approach for controllable image-to-video generation. Our method can control object motion and camera dynamics for arbitrary input images and any number of objects or regions of a scene.

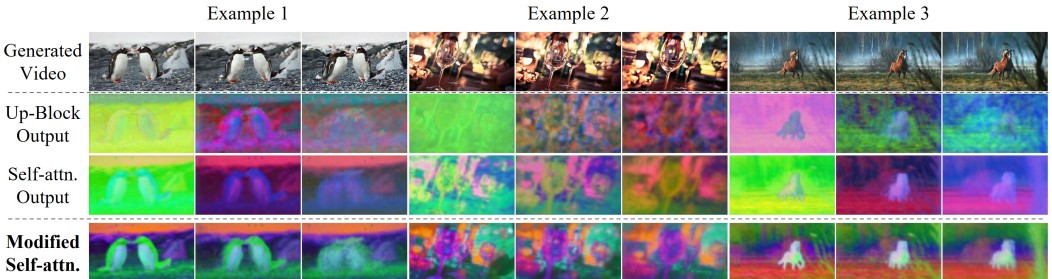

Figure 2: **Semantic correspondences in video diffusion models.** We analyze feature maps in the image-to-video diffusion model SVD (Blattmann et al., 2023a) for three generated video sequences (row 1). We use PCA to visualize the features at diffusion timestep 30 (out of 50) at the output of an upsampling block (row 2), a self-attention layer (row 3), and the same self-attention layer after our alignment procedure (row 4). Although output feature maps of upsampling blocks in image diffusion models are known to encode semantic information (Tang et al., 2023), we only observe weak semantic correspondences across frames in SVD. Thus, we focus on the self-attention layer and modify it to produce feature maps that are semantically aligned across frames.

- We conduct extensive experiments to show superior performance over zero-shot baselines while significantly narrowing down the performance gap with supervised baselines in visual and motion quality.

## 2 RELATED WORK

**Diffusion-based image-to-video generation.** With the recent advances in diffusion models (Sohl-Dickstein et al., 2015; Ho et al., 2020), image animation has achieved tremendous progress (Wang et al., 2023; Chen et al., 2023; 2024a; Guo et al., 2024). Early methods inflate pre-trained text-to-image models (Rombach et al., 2022) to add motions to an image (Wu et al., 2023; Khacha-tryan et al., 2023; Singer et al., 2023). A notable example is AnimateDiff (Guo et al., 2024), which learns low-rank adapters (Hu et al., 2022) for different motions. Later works seek to inject a conditioning frame into a pre-trained text-to-video model (Ho et al., 2022b;a; He et al., 2022). VideoCrafter1 (Chen et al., 2023) leverages a dual cross-attention layer to condition on features of both the image and the text prompt. DynamicCrafter (Xing et al., 2024) further improves it by concatenating the input image with noisy latent. Stable Video Diffusion (SVD) (Blattmann et al., 2023a) instead works in an image-only manner, which replaces the CLIP (Radford et al., 2021) text embedding of the text prompt with the CLIP image embedding of the conditioning frame. However, none of these models support direct trajectory control of scene elements, and rather require multiple attempts to obtain a desired result. In this work, we aim to enable intuitive motion control in animating a pre-existing image. Since SVD only takes in an image without any text prompt, we utilize it as the base model following prior work (Yin et al., 2023; Wu et al., 2024c).

**Spatial control in image diffusion models.** One common way to incorporate spatial control into the image generation process is to fine-tune pre-trained models to incorporate conditioning on depth maps or bounding boxes (Zhang et al., 2023; Ye et al., 2023; Avrahami et al., 2023; Li et al., 2023; Goel et al., 2024; Wang et al., 2024b). While these methods demonstrate high fidelity, they require excessive computing resources and labor-intensive data annotations. Therefore, several tuning-free approaches have been proposed (Cao et al., 2023; Chen et al., 2024b; Feng et al., 2023; Hertz et al., 2023). Self-Guidance (Epstein et al., 2023) Attend-and-Excite (Chefer et al., 2023) , and TraDiffusion (Wu et al., 2024a) control the image layout by manipulating intermediate attention maps. They first estimate the generated objects' positions using the attention maps produced from text–image cross-attention layers and then optimize the latent to increase the attention values at specific positions. However, these approaches require associating the control target with a specific token in the text prompt and thus cannot be directly applied to image-only editing. Closest to ours are methods that alter image layouts without text input (Pan et al., 2023; Mou et al., 2024a;b; Ling et al., 2023; Liu et al., 2024; Zhang et al., 2024c; Cui et al., 2024; Hou et al., 2024b; Zhao et al., 2024; Shi et al., 2024). They enforce similarity between semantically correlated feature maps extracted from

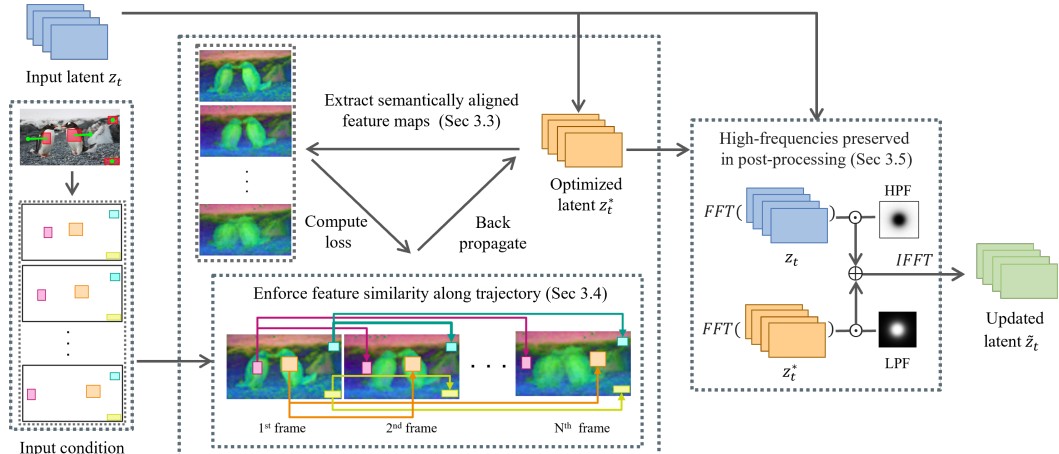

Figure 3: **Overview of the controllable image-to-video generation framework.** To control trajectories of scene elements, we optimize the latent $z_t$ at specific denoising timesteps $t$ of a pre-trained video diffusion model. First, we extract semantically aligned feature maps from the denoising U-Net to estimate the video layout. Next, we enforce cross-frame feature similarity along the bounding box trajectory to drive the motion of each region. To preserve the visual quality of the generated video, a frequency-based post-processing method is applied to retain high-frequency noise of the original latent $z_t$. The updated latent $\tilde{z}_t$ is input to the next denoising step.

the upsampling blocks of a denoising U-Net (Tang et al., 2023; Namekata et al., 2024; Hedlin et al., 2024; Zhang et al., 2024a; Luo et al., 2023). In contrast, we show that the cross-frame semantic correspondence of upsampling feature maps in image-to-video diffusion models (Blattmann et al., 2023a) is weak. Thus, optimization directly based on these feature maps leads to sub-optimal results.

**Motion control in video diffusion models.** Several recent works have studied camera pose control in video diffusion models (He et al., 2024a; Xu et al., 2024a; Kuang et al., 2024; Hu et al., 2024; Xiao et al., 2024b; Hou et al., 2024a; Bahmani et al., 2024; Li et al., 2024; Zhang et al., 2024b). The representative work MotionCtrl (Wang et al., 2024c) fine-tunes pre-trained video generators to follow camera trajectory input. However, video datasets with accurate camera pose annotations are limited (Zhou et al., 2018), and fine-tuning video models requires high computation costs. Another line of work focuses on object trajectory control (Wang et al., 2023; Yin et al., 2023; Wu et al., 2024c; Zhou et al., 2024; Wang et al., 2024a; Zhang et al., 2024d), among which our work is particularly related to the tuning-free variants (Qiu et al., 2024; Ma et al., 2023; Yang et al., 2024; Jain et al., 2024; Yu et al., 2024). Yet, all of these methods focus on text-based generation. In contrast, our method uses an image-to-video model and thus can turn existing, real-world images into controllable videos. Moreover, we can control camera motion by specifying trajectories of background regions.

## 3 METHOD

In this section, we describe our method for the trajectory control task in image-to-video generation (Sec. 3.1). Our framework, SG-I2V, builds on the publicly available image-to-video diffusion model Stable Video Diffusion (SVD) (Blattmann et al., 2023a), and consists of two main steps. First, we extract and semantically align the feature maps from a specific layer of SVD during the early steps of the diffusion process (Sec. 3.2); we show that such feature maps are especially effective at influencing motion in the output video. Second, we optimize the noisy latent (i.e., the input to the denoising network) to enforce similarity between features within the bounding box trajectories (Sec. 3.3). However, we find that naive optimization of latent is prone to overfitting and often results in low-quality generation. Thus, we employ frequency-based post-processing to retain an in-distribution noisy latent (Sec. 3.4). Our entire pipeline is summarized in Fig. 3.

## 3.1 TRAJECTORY CONTROL IN IMAGE-TO-VIDEO GENERATION

The goal of our work is to build a zero-shot framework that takes an input image $\mathbf{I} \in \mathbb{R}^{H \times W \times 3}$ (with height $H$, width $W$) and generates a video with $N$ frames, where elements in the input image move in a user-specified fashion. Inspired by DragAnything (Wu et al., 2024c), we further assume that a user provides a set of $B$ input bounding box trajectories $\{\mathcal{B}_b\}_{b=1}^{B}$, each parameterized by a height $h_b$ and width $w_b$, as well as center point coordinates for each output video frame: $\mathbf{c}_{b,n} \in \mathbb{R}^2, 1 \leq n \leq N$. For simplicity, we assume bounding boxes cannot extend outside the image, and we denote the $b$-th bounding box in the $n$-th frame as $\mathcal{B}_{b,n} = \{h_b, w_b, \mathbf{c}_{b,n}\}$.

Our aim is to constrain regions of the input image falling within a bounding box to follow the trajectory of the same bounding box in the output video. Thus, the motion of dynamic foreground objects can be controlled by placing bounding boxes around them and specifying the desired trajectory. On the other hand, camera motion can be specified by placing bounding boxes on static background regions and specifying a trajectory opposite to the desired camera movement. Further, we can also set the bounding box trajectory to the zero vector to keep the region static. Overall, this formulation provides intuitive and unified control over object and camera motion, which are sometimes treated separately in previous controllable image-to-video frameworks (Wang et al., 2024c; Yang et al., 2024; Li et al., 2024).

## 3.2 EXTRACTING SEMANTIC VIDEO LAYOUT

**Preliminaries: Stable Video Diffusion.** Video diffusion models (Ho et al., 2022b) learn a data distribution $\boldsymbol{x}_0 \sim p_\theta(\boldsymbol{x}_0)$ by gradually denoising a video corrupted by Gaussian noise. The output denoised video is thus drawn from the distribution $p_\theta(\boldsymbol{x}_0) = \int p_\theta(\boldsymbol{x}_{0:T}) \, d\boldsymbol{x}_{1:T}$, where $\boldsymbol{x}_0 \in \mathbb{R}^{N \times H \times W}$ is a clean video, and $\boldsymbol{x}_{1:T}$ are intermediate noisy samples. For simplicity, we omit the channel dimension throughout the paper. To reduce computation, Stable Video Diffusion (SVD) (Blattmann et al., 2023a) performs the diffusion process in a latent space, where a variational autoencoder (Kingma & Welling, 2013) maps a raw video $\boldsymbol{x}_0$ to a latent $\boldsymbol{z}_0 \in \mathbb{R}^{N \times h \times w}$.
Since this work aims to animate an existing image, we utilize the image-to-video variant of SVD, which concatenates a conditioning frame with noisy latent ($\boldsymbol{z}_t$) and runs a 3D U-Net (Ronneberger et al., 2015) to predict the noise. The 3D U-Net contains a downsampling and an upsampling path. Specifically, the upsampling path consists of three stages operating at different resolutions, where each stage contains three blocks with interleaved residual blocks (He et al., 2016), spatial, and temporal attention layers (Vaswani et al., 2017). We will call these three stages bottom, middle, and top from lower to higher resolution. For more details, we refer readers to the original paper of SVD (Blattmann et al., 2023a).

**SVD feature map analysis.** In image diffusion models, prior work has shown that output feature maps of upsampling blocks in the middle stage of the denoising U-Net are *semantically aligned* (Tang et al., 2023; Namekata et al., 2024; Hedlin et al., 2024; Zhang et al., 2024a; Luo et al., 2023), i.e., regions belonging to the same object tend to have similar feature vectors. Such semantically aligned feature maps are useful in estimating the layout of generated images, enabling spatial control of objects (Shi et al., 2024; Mou et al., 2024a). Therefore, we first examine whether SVD feature maps are also semantically correlated across *both spatial and temporal dimension*. Fig. 2 visualizes the principal components of feature maps extracted from the upsampling block and spatial attention layers. We observe that SVD feature maps exhibit weak semantic correspondence across frames at early denoising steps, leading to inaccurate object trajectory estimation. Yet, we want to operate at early steps as they decide the structure of generated videos (Materzynska et al., 2023). This dilemma prompts us to align these features before applying optimization.

**Feature alignment with modified self-attention.** SVD leverages separate spatial and temporal self-attention to model the entire video. Since spatial self-attention is only applied per frame, it does not produce cross-frame aligned features. While temporal attention communicates across frames, it only attends to the same pixel position on the feature map, which may be inadequate for capturing semantic information spatially. To address this issue, inspired by (Wu et al., 2023), we modify the spatial self-attention on each frame to directly attend to the first frame. Concretely, for the $n$-th frame, the original spatial self-attention works as $\boldsymbol{F}_n = \text{Softmax}(\frac{\boldsymbol{Q}_n \cdot \boldsymbol{K}_n^T}{\sqrt{D}}) \cdot \boldsymbol{V}_n$, where $\boldsymbol{F}_n$ is the outputs of self-attention, $\boldsymbol{Q}, \boldsymbol{K}, \boldsymbol{V}$ are the query, key, and value tokens, respectively, and

$D$ is the dimensionality of the key and query tokens Vaswani et al. (2017). Instead, we replace the key $K_n$ and value $V_n$ of each frame with $K_1$ and $V_1$ from the first frame, leading to a new operation $\tilde{F}_n = \text{SoftMax}\left(\frac{Q_n \cdot \text{SG}(K_1)^T}{\sqrt{D}}\right) \cdot \text{SG}(V_1)$. We apply a stop gradient $\text{SG}(\cdot)$ on $K_1$ and $V_1$ to stabilize the subsequent optimization process. Now, all the modified feature maps $\tilde{F}_n$ are weighted combinations of $V_1$, exhibiting a stronger cross-frame correspondence while still maintaining the object layout of each frame, as shown in the bottom row of Fig. 2. Notably, this modification occurs during the loss computation only and does not affect the denoising steps.

## 3.3 TRAJECTORY CONTROL WITH LATENT OPTIMIZATION

So far, we have obtained spatio-temporally aligned feature maps $\tilde{F}_n(z_t) \in \mathbb{R}^{h \times w \times d}$ at each frame given the noisy latent $z_t$ as input (for simplicity, we resize $\tilde{F}_n$ to the same resolution as the noisy latent). Recall that our goal is to control the output video frames so that the bounding boxes $\mathcal{B}_b$ identified in the first frame move along the associated trajectories. Inspired by prior drag-based control methods (Shi et al., 2024; Pan et al., 2023), we optimize the noisy latent $z_t$ to enforce cross-frame similarity between features within bounding boxes. The optimization objective is as follows:

$$z_t^* = \arg\min_{z_t} \sum_{b \in [1,B], n \in [2,N]} \|G_b \odot \left(\tilde{F}_n(z_t)[\mathcal{B}_{b,n}] - \text{SG}(\tilde{F}_1(z_t)[\mathcal{B}_{b,1}])\right)\|_2, \qquad (1)$$

where $\odot$ is Hadamard product, and $\tilde{F}_n(z_t)[\mathcal{B}_{b,n}] \in \mathbb{R}^{h_b \times w_b \times d}$ is feature maps cropped by the bounding box $\mathcal{B}_{b,n}$. Following DragAnything (Wu et al., 2024c), we weight the feature difference using a Gaussian heatmap $G_b \in \mathbb{R}^{h \times w}$. This focuses on optimizing pixels closer to the bounding box center, as pixels near the edge may be background pixels that we do not want to move.

**Selective latent optimization.** We optimize Eq. (1) on a subset of denoising timesteps and self-attention layers. Concretely, we only select early denoising timesteps as the coarse structure of output frames is determined at these timesteps (Wang & Vastola, 2023; Materzynska et al., 2023). In addition, consistent with previous works in image diffusion models (Shi et al., 2024; Mou et al., 2024a), we observe that feature maps extracted from the middle stage of the denoising U-Net are more semantically correlated, and thus we use them for optimization.

## 3.4 HIGH-FREQUENCY PRESERVED POST-PROCESSING

Although the presented pipeline already enables trajectory control in video generation, we notice a quality degradation in generated videos. We attribute this degradation to the deviation of $z_t^*$ from the sampling distribution of the diffusion process after optimization. A recent work FreeInit (Wu et al., 2024b) observed that motions of generated videos are mostly encoded in the low-frequency component of noisy latent. Inspired by this, we propose to discard the high-frequency component of the optimized latent $z_t^*$ and replace it with the high-frequency component of the original latent $z_t$. Formally, we obtain the new latent $\tilde{z}_t$ as follows:

$$\tilde{z}_t = \text{IFFT}_{2D}\left(\text{FFT}_{2D}(z_t^*) \odot \mathbf{H}_\gamma + \text{FFT}_{2D}(z_t) \odot (1 - \mathbf{H}_\gamma)\right), \qquad (2)$$

where $\text{FFT}_{2D}$ is the Fast Fourier transformation applied to each frame, and $\text{IFFT}_{2D}$ is the corresponding inverse operation. We set $\mathbf{H}_\gamma$ as the frequency response of a 2D low-pass filter (we follow FreeTraj and use a Butterworth filter) with cut-off frequency $\gamma$. This post-processing step retains the target motion signals encoded in the low-frequency component of $z_t^*$ while eliminating undesirable high-frequency disruptions.

## 4 EXPERIMENTS

### 4.1 EXPERIMENTAL SETUP

**Implementation details.** In all experiments, we leverage the image-to-video variant of Stable Video Diffusion (Blattmann et al., 2023a) to generate videos with 14 frames and $576 \times 1024$ resolution. The default discrete Euler scheduler (Karras et al., 2022) is applied with $T = 50$ sampling steps. We extract feature maps from the last two self-attention layers from the middle stage in the denoising U-Net. We optimize Eq. (1) at the early denoising timesteps $t \in [45, 44, ..., 30]$ for 5 iterations per timestep. We use the AdamW optimizer (Loshchilov & Hutter, 2019) with a learning

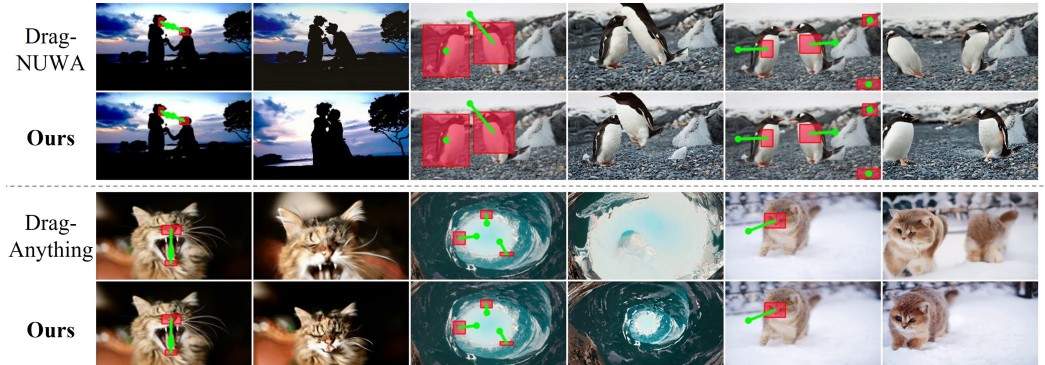

Figure 4: **Failure cases in supervised baselines.** We observe that DragNUWA tends to distort objects rather than move them, and DragAnything is weak at part-level control as it is designed for entity-level control. In contrast, our method can generate videos with natural motion for diverse object and camera trajectories. Please see *our project page* for additional comparisons.

rate of $0.21$. All these design choices are carefully ablated and analyzed in Sec. 4.4. During loss calculation, Gaussian heatmap $\boldsymbol{G}_b$ is constructed following (Wu et al., 2024c), where a heatmap for a bounding box of size $(h_b, w_b)$ is created by Gaussian distribution with standard deviation $\sigma = (0.2h_b, 0.2w_b)$. For the low-pass filter $\mathbf{H}_\gamma$, we set the cut-off frequency $\gamma$ to $0.5$.

**Baselines.** We compare with methods that enable motion control on existing images and have publicly available code. Specifically, we compare with supervised baselines *DragNUWA* (Yin et al., 2023) and *DragAnything* (Wu et al., 2024c), which both add motion adapters to SVD. We also compare with *Image Conductor* (Li et al., 2024), which is based on AnimateDiff (Guo et al., 2024). Since all supervised baselines are fine-tuned to generate videos at a lower resolution, we resize the generated videos from all the methods to $320 \times 576$ for a fair comparison. Since no previous methods exist for zero-shot trajectory-controlled image-to-video generation, we adopt techniques from text-to-video methods to create new baselines. Specifically, *FreeTraj*[†] incorporates the noise initialization technique from (Qiu et al., 2024) by copy-pasting the initial noise on the first frame to other frames along the trajectories. *DragDiffusion*[†] is inspired by the image editing method in (Shi et al., 2024), which utilizes feature maps extracted from the outputs of upsampling blocks to guide the generation process without feature alignment. *MOFT*[†] instead utilizes feature maps derived from *Content Correlation Removal* proposed by Xiao et al. (2024a) known to encode motion information.

**Datasets and evaluation metrics.** Following prior works (Wu et al., 2024c; Zhou et al., 2024), we evaluate our method on the validation set of the VIPSeg dataset (Miao et al., 2022). We test on the same control regions and target trajectories as DragAnything, where the size of our bounding boxes is the same as the diameter of the circles in their work. For quantitative metrics, we report Frechet Inception Distance (FID) (Heusel et al., 2017) and Frechet Video Distance (FVD) (Unterthiner et al., 2018) to measure the visual quality, and ObjMC (Wu et al., 2024c) to measure the motion fidelity. ObjMC computes the average distance between generated and target trajectories, where Co-Tracker (Karaev et al., 2024) is used to estimate the trajectory of generated videos.

## 4.2 QUALITATIVE RESULTS

Fig. 1 presents the versatile control ability of our method. We can control foreground objects to perform rigid motions, such as trains moving, and non-rigid motions, such as the movement of human hairs. In addition, we can control non-physical entities such as smoke and fire. The moved scene elements naturally adapt to the new location while preserving their original identity. Thanks to our general formulation of trajectories, camera motion control is also supported. We highly encourage the reader to view *our project page* for additional results.

## 4.3 QUANTITATIVE ASSESSMENT

**Comparison with supervised baselines.** We provide quantitative comparisons to baselines in Tab. 1. Despite being trained on large-scale datasets, Image Conductor underperforms our method

| Method | FID (↓) | FVD (↓) | ObjMC (↓) | Zero-shot | Resolution | Backbone |
|--------|---------|---------|-----------|-----------|------------|----------|
| Image Conductor | 48.81 | 463.21 | 21.07 | | $256 \times 384$ | AnimateDiff v3 |
| DragNUWA v1.5 | 30.73 | 253.57 | 10.84 | | $320 \times 576$ | SVD |
| DragAnything | 30.81 | 268.47 | 11.64 | | $320 \times 576$ | SVD |
| SVD (No Control) | 30.50 | 340.52 | 39.59 | ✓ | $576 \times 1024$ | SVD |
| FreeTraj[†] | 46.61 | 394.14 | 36.43 | ✓ | $576 \times 1024$ | SVD |
| MOFT[†] | 30.76 | 402.09 | 33.58 | ✓ | $576 \times 1024$ | SVD |
| DragDiffusion[†] | 30.93 | 458.29 | 31.49 | ✓ | $576 \times 1024$ | SVD |
| **SG-I2V** | 28.87 | 298.10 | 14.43 | ✓ | $576 \times 1024$ | SVD |

[†] indicates methods adapted to our image-to-video setting.

Table 1: **Quantitative comparison on the VIPSeg dataset.** Despite being a zero-shot method, we achieve small gaps in motion fidelity (ObjMC) to supervised baselines without degrading video quality (FID, FVD). Furthermore, our approach outperforms zero-shot baselines across all metrics.

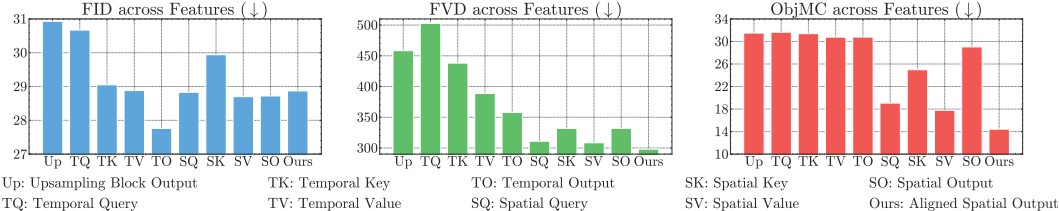

Figure 5: **Performance across U-Net feature maps used to compute loss in Eq. (1).** For all metrics, lower values are better. *Temporal* and *spatial* refer to the temporal and spatial self-attention layers. We find that features extracted from self-attention layers generally perform better than those from upsampling blocks and temporal attention layers. In addition, using the feature maps of our modified self-attention layer achieves the best results, since they are semantically aligned across frames. Corresponding qualitative visuals are presented in Fig. 13 and *our project page*.

by a large margin, mainly because of the limited capacity of the base model AnimateDiff. Compared to methods that also build upon SVD, we achieve competitive performance in visual quality, with slightly worse motion fidelity. Yet, these methods are trained to generate low-resolution (i.e., $320 \times 576$) videos due to the high cost of fine-tuning at high resolution. At the same time, our tuning-free approach can maintain the original resolution of SVD (i.e., $576 \times 1024$). Fig. 4 illustrates comparison in failure cases of supervised baselines. Similar to observations in (Wu et al., 2024c), DragNUWA tends to distort objects rather than naturally move them, while our method can generate more natural movements. DragAnything is weak at part-level motion control (e.g., closing a cat's mouth) as it is only trained on datasets annotated with entity-level control, such as object segmentation masks. In contrast, our approach can handle different granularities of control regions.

**Comparison with adapted zero-shot baselines.** The noise initialization technique in FreeTraj[†] improves motion control slightly compared to original videos but significantly degrades visual quality. This indicates that motion prior can not be easily incorporated into initial noises of SVD by a hand-crafted algorithm. DragDiffusion[†] and MOFT[†] leverages feature maps that are not semantically corresponding across frames. As a result, the motion fidelity is much lower than ours. Overall, SG-I2V significantly outperforms zero-shot baselines across all metrics. This highlights that zero-shot techniques applied in text-to-video diffusion models do not necessarily transfer to image-to-video models.

## 4.4 ABLATION STUDIES

**Feature map selection.** In Fig. 5, we analyze the effect of the choice of U-Net feature map in the optimization of Eq. (1). DIFT (Tang et al., 2023) pointed out that outputs from the U-Net upsampling block in image diffusion models have strong semantic correspondence across *spatial* dimensions, which is used in image editing methods (Shi et al., 2024). However, we find them to have inferior correspondence across the *temporal* dimension, and optimizing them leads to inaccurate object trajectories, as indicated by the high ObjMC value. Next, we examine features in temporal

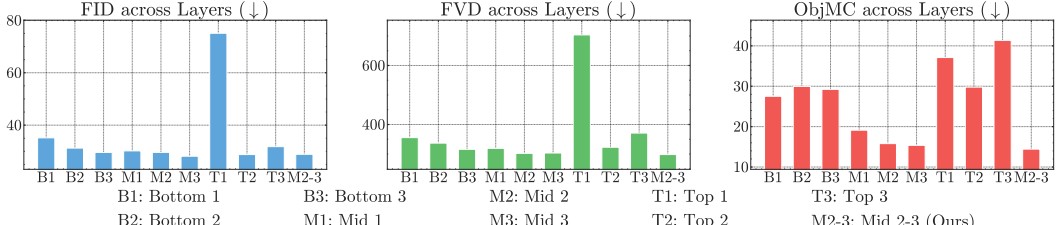

Figure 6: **Performance across U-Net layers used to extract feature maps.** Lower is better for all metrics. *Bottom*, *mid*, and *top* indicate the three resolution levels in the U-Net's upsampling path, each containing three self-attention layers numbered 1, 2, and 3. for example "M2-3" means applying the loss to features from both mid-resolution layers 2 and 3. We observe that mid-resolution feature maps perform best for trajectory guidance. In addition, using features from both M2 and M3 leads to the best result. See *our project page* for visualizations.

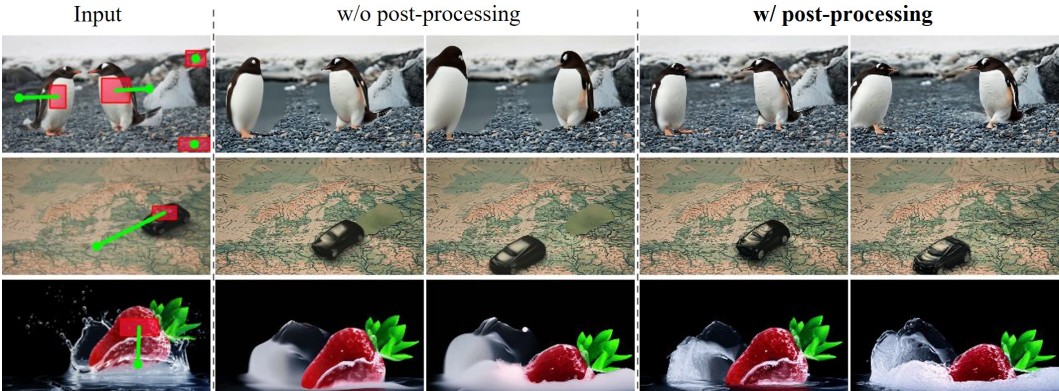

Figure 7: **Effect of high-frequency preservation in post-processing.** Videos without post-processing tend to demonstrate oversmoothing and have artifacts. In contrast, our post-processing technique retains videos with sharp details and eliminates most of the artifacts. See *our project page* for more examples.

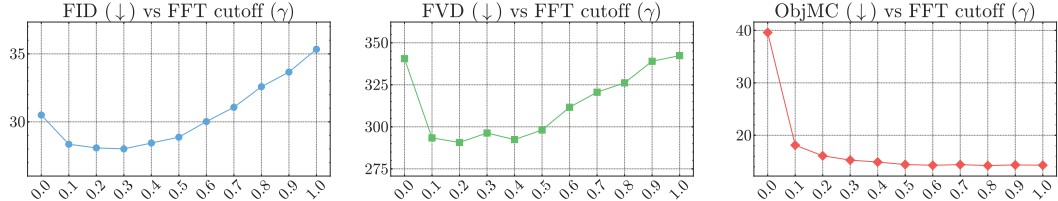

Figure 8: **Study of the cut-off frequency in post-processing.** Lower is better for all metrics. The value $\gamma$ indicates the cut-off frequency. Fully keeping the optimized latent ($\gamma = 1$) results in degraded video quality, as shown by high FID and FVD values. On the other hand, replacing too many frequency components diminishes motion control, as indicated by the increasing ObjMC.

self-attention layers. However, using these features also leads to inferior motion guidance, as indicated by the high ObjMC errors. This may be because the temporal layers in SVD always attend to the same spatial location, thus focusing less on each frame's spatial layout. Finally, we study each component in spatial self-attention layers. As discussed in Sec. 3.2, the lack of cross-frame correspondence in the original self-attention is problematic—in Fig. 5, we see that this results in worse ObjMC and FVD scores. Overall, optimizing outputs from our modified self-attention operation achieves the best motion fidelity, showing the importance of feature alignment.

**Cut-off frequency in post-processing.** We study the cut-off frequency in our high-frequency-preserving post-processing step in Fig. 8. Naively keeping the optimized latent ($\gamma = 1$) degrades the visual quality drastically, as shown by the higher FID and FVD. Yet, discarding part of the high-frequency component has negligible impact on motion control while eliminating most artifacts. We thus choose $\gamma = 0.5$ as a sweet spot for effective video quality restoration.

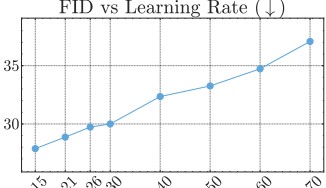 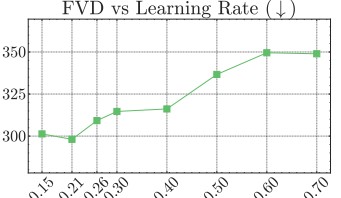 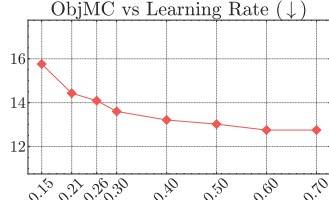

Figure 9: **Ablation on optimization learning rates.** Larger learning rates lead to video quality degradation (i.e., higher FID and FVD), while smaller learning rates result in lower motion fidelity (i.e., higher ObjMC). We choose the learning rate considering this tradeoff.

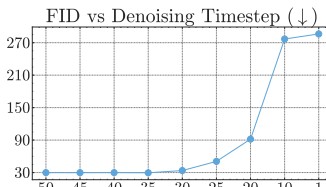 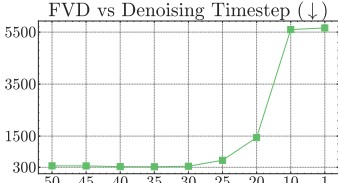 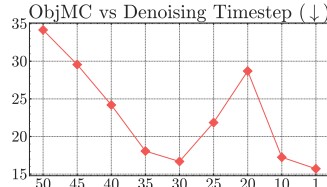

Figure 10: **Effect of optimizing latent at individual denoising timesteps.** For all metrics, lower values are better. Here, we optimize Eq. (1) on a single denoising timestep ($t = 50$ corresponds to standard Gaussian noise), and we find middle timesteps (e.g. $t = 30$) achieve the best motion fidelity while maintaining visual quality. More results on optimizing the latent at multiple timesteps can be found in Fig. 16. See Fig. 15 and *our project page* for qualitative comparisons.

**U-Net layer.** Fig. 6 ablates from which layer to extract the feature maps. The upsampling path of SVD's U-Net contains three stages (bottom, mid, top), each with three spatial self-attention layers (indexed 1, 2, 3). We observe that blocks at the middle-resolution level capture most semantically meaningful information, which aligns with prior work in image diffusion models (Shi et al., 2024; Tang et al., 2023). We also try joint optimization of several layers, which gives the best performance.

**Learning rate.** Fig. 9 examines the effect of learning rate under a fixed number of optimization steps. We observe a clear trade-off between visual quality and motion fidelity of generated videos. This is because a large learning rate quickly leads to noisy latents that are out of distribution.

**Denoising timesteps.** Fig. 10 summarizes the effects of optimizing features across different denoising timesteps ($t = 50$ corresponds to standard Gaussian noise). Lower timesteps (e.g., $t = 20, 1$) significantly impair visual quality, motivating us to optimize only on earlier denoising steps. Conversely, timesteps for $t > 45$ degrade motion fidelity due to the lack of detailed semantic information at extremely high noise levels. Notably, ObjMC improves at $t \in [20, 1]$, but due to the Co-tracker tracking moving artifacts *(e.g., see project webpage)*. Based on these observations, we optimize feature maps extracted between timesteps 30–45 to balance visual quality and motion fidelity.

## 5 CONCLUSION

Our work introduces the first framework for zero-shot trajectory control in image-to-video generation. Our thorough analysis of diffusion features reveals that the knowledge acquired in the pre-trained image-to-video diffusion models can guide them to generate videos with desired motions. Quantitative and qualitative results demonstrate the effectiveness of our approach, both on synthetic and real-world images. We hope our findings can shed light on the inner mechanism of image-to-video diffusion models and inspire better architecture designs in the future.

**Limitations and future work.** Since our pipeline works in a zero-shot and self-guided manner, the base video diffusion model bounds the quality of generated videos, e.g., for subjects with large motion or complex physical interactions (Blattmann et al., 2023a). Another area for improvement is the out-of-distribution issue when optimizing the latent code. Although we have incorporated frequency-based post-processing to mitigate it, we sometimes still observe artifacts when we set the learning rate higher. How to alter the denoising process while keeping in-distribution latents is an open problem itself (He et al., 2024b; Garibi et al., 2024). With the recent rapid progress in video generators (Brooks et al., 2024; Gupta et al., 2023), we also expect our framework could be extended to newly released models to achieve improved generation quality.

ACKNOWLEDGEMENTS

DBL acknowledges support from NSERC under the RGPIN and Alliance programs, the Canada Foundation for Innovation, and the Ontario Research Fund.

ETHICS STATEMENT

Currently, our synthesized videos are not yet indistinguishable from real footage. Still, we anticipate that future iterations of large video models will achieve even greater quality, potentially producing videos that look photoreal. This prospect brings significant ethical and safety considerations, as cutting-edge generative models can be misused to ill effect. We advocate against misuse of generative models to malign or misinform, and we encourage safe and responsible use of such models (Google, 2023).

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

## A    Experimental Setup Details

**Details of FVD.** As mentioned in Sec. 4.1, we resize generated videos to $320 \times 576$ before feeding them into the evaluation scripts to ensure a fair comparison. For FVD computation, we further resize all videos to $256 \times 256$, following DragAnything (Wu et al., 2024c).

**Details of ObjMC.** Following DragAnything (Wu et al., 2024c), we compute ObjMC to evaluate motion fidelity. ObjMC is defined as the framewise average distance between the generated and ground truth trajectories, where the ground truth trajectories are reused from DragAnything, and the generated trajectories are estimated using Co-Tracker v2 (Karaev et al., 2024). Since the VIPSeg dataset contains videos with varying resolutions (e.g., $480 \times 800$, $1440 \times 2560$), we resize all videos to a uniform resolution (as specified in Tab. 1) before feeding them into the models.

Unlike DragAnything, which resizes generated videos back to their original resolutions for ObjMC computation, we resize all videos to a fixed resolution of $320 \times 576$. Additionally, we exclude ground truth trajectory points that fall outside the image space due to objects moving out of frame. We also omit short videos with fewer than 14 frames from the evaluation.

**Details of zero-shot adopted methods.** Due to the unavailability of zero-shot trajectory-controlled image-to-video generation methods, we adopt techniques from text-to-image and text-to-video methods to establish new baselines in Tab. 1.

*FreeTraj*[†] incorporates the noise initialization technique (called *trajectory injection*) from (Qiu et al., 2024). This work has shown that introducing motion bias into the initial noise—by copy-pasting the noise from the first frame to other frames along the trajectory—can influence motion in the generated videos of text-to-video diffusion models. However, we did not observe similar effects in image-to-video diffusion models. Instead, we found that this approach degraded visual quality, likely because the modified input latents fell out of distribution.

*DragDiffusion*[†] is inspired by the image editing technique in (Shi et al., 2024), which utilizes feature maps extracted from the outputs of upsampling blocks to guide the generation process. Specifically, we optimize on the feature maps output by the second and third upsampling blocks at the middle resolution level. Although the original DragDiffusion does not include high-frequency preserved post-processing, we incorporate it in our baseline to enhance visual quality. We use the same hyper-parameters as in our method.

*MOFT*[†] is similar to *DragDiffusion*[†] but optimizes feature maps derived from *Content Correlation Removal*, as proposed by Xiao et al. (2024a). Specifically, we optimize on the outputs of the upsampling layers subtracted by their mean features averaged across frames. These feature maps are well-suited for reference-guided motion generation because they are motion-consistent (i.e., pixels with similar motion exhibit similar feature vectors), enabling direct comparison with reference motion feature maps. However, they are less sensitive to semantic information and, therefore, ineffective for our optimization, where reference motion is unavailable.

## B    Additional Qualitative Results

We strongly encourage readers to refer to our project website: `https://kmcode1.github.io/Projects/SG-I2V` for more visualizations in video format, where we have released qualitative results with various trajectories, qualitative results on VIPSeg dataset, qualitative comparisons with supervised and zero-shot baselines, and qualitative analysis for ablation study.

## C    Additional Results for Ablation Study

In this subsection, we provide full results for our ablation analysis.

**Inference time.** Our method generates 14 frames videos at a resolution of $576 \times 1024$ with $T = 50$ sampling steps. Optimization is performed from the timestep 45 to the timestep 30 with 5 iterations per timestep, totaling 75 optimization iterations. The runtime depends on the number of trajectory conditions, with an average runtime of 305 seconds on the VIPSeg dataset with A6000 48GB. Due to the need of backpropagation, the peak GPU memory usage amounts to around 30 GB.

| Mask shape | FID ($\downarrow$) | FVD ($\downarrow$) | ObjMC ($\downarrow$) |
|---|---|---|---|
| Gaussian weighting | 28.87 | 298.10 | 14.43 |
| Identity weighting | 28.88 | 300.20 | 14.72 |

Table 2: **Ablation on Gaussian weighting on the VIPSeg dataset.** Using a Gaussian heatmap in loss computation consistently improves the results across all metrics.

**PCA visualization.** Fig. 11 and Fig. 12 present the visual results of our PCA analysis on feature maps extracted from various layers of SVD across different timesteps (corresponding to Sec. 3.2). We can confirm that our modified self-attention consistently produces semantically aligned feature maps throughout the denoising process.

**Feature map selection.** Fig. 13 presents the visual results of ablating feature maps for optimization. Performing optimization with feature maps naively extracted from original self/temporal attention or upsampling blocks fails to follow the input trajectory due to the weak semantic alignment across frames. In contrast, performing optimization with our modified self-attention features successfully produces videos consistent with the input trajectory. This highlights the importance of extracting semantically aligned feature maps for our optimization.

**U-Net layer.** Consistent with the quantitative results (Fig. 6), Fig. 14 has qualitatively confirmed that optimizing with feature maps extracted from the middle layers of the upward path produces plausible videos consistent with the input trajectory.

**Denoising timesteps.** Continuing from the main paper, we further analyze the effect of extracting feature maps from different timesteps. Fig. 15 qualitatively demonstrates the effect of optimizing latent at individual denoising timesteps. Consistent with the quantitative results demonstrated in Fig. 10, performing optimization at later stage of denoising steps (i.e. $t = 10, 20$) severely introduces artifacts in the generated videos. Next, we examine the effects of performing optimization for multiple timesteps by fixing the total number of iterations for optimization. As summarized in Fig. 16, we observe similar trends as optimizing individual timesteps, where performing optimization at the later stage of the denoising process significantly degrades visual quality, while performing optimization up to timestep $40$ is not enough for motion control. Overall, running optimization for multiple timesteps performs better than running optimization only at a single timestep, and hence we adapt continuous timesteps for our experiments.

**Gaussian weighting.** Following DragAnything, we weigh the feature difference using a Gaussian heatmap during the loss computation in Sec. 3.3. This accounts for potential errors in placing bounding boxes, where the bounding box may include background pixels around the object that are not intended to be moved. As shown in our ablation on the VIPSeg dataset (Tab. 2), Gaussian weighting results in small but consistent improvement across all metrics.

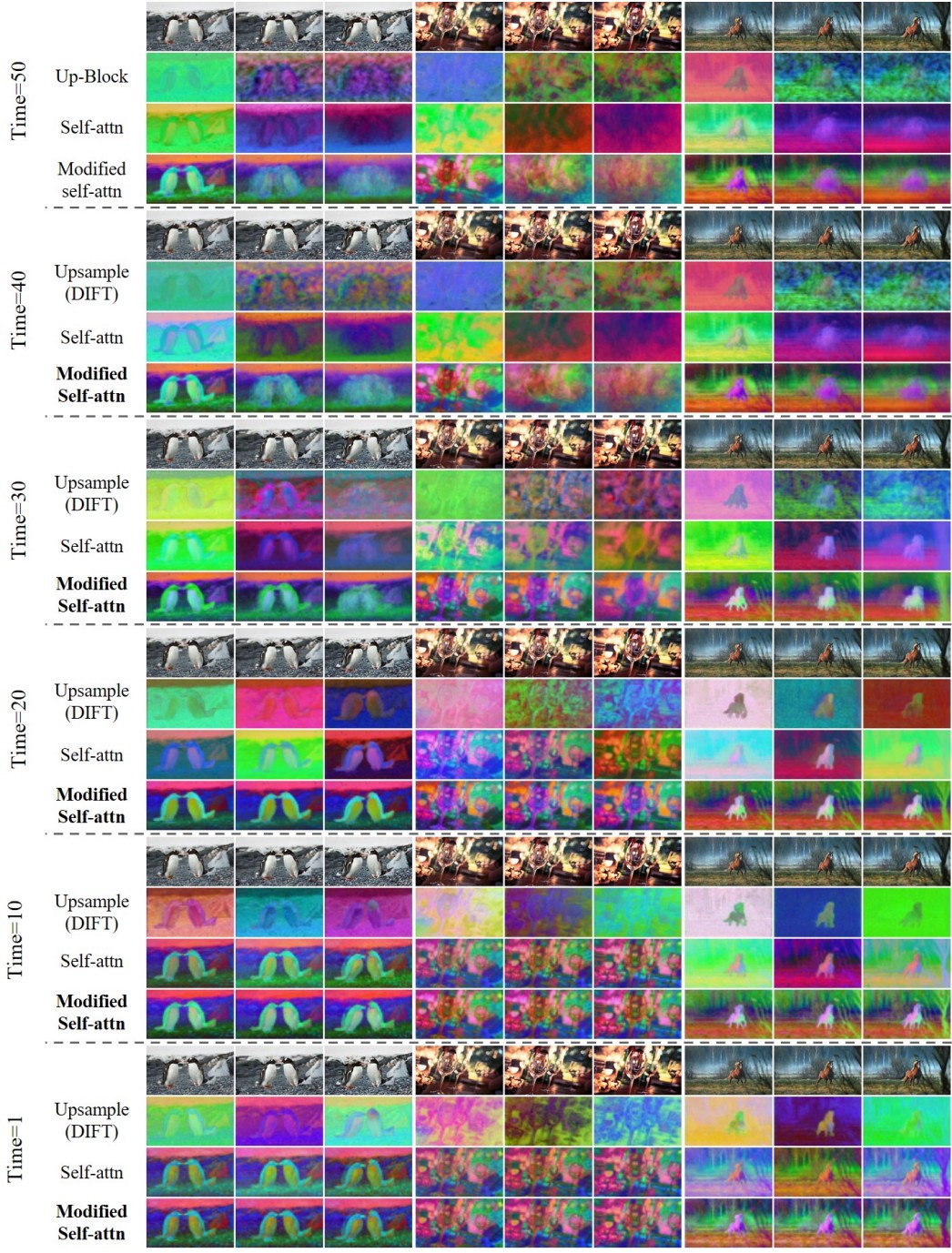

Figure 11: **Semantic correspondences in video diffusion models across timesteps.** Output feature maps of upsampling blocks have limited semantic correspondences across frames. In contrast, our modified self-attention layers produce semantically aligned feature maps across all the timesteps.

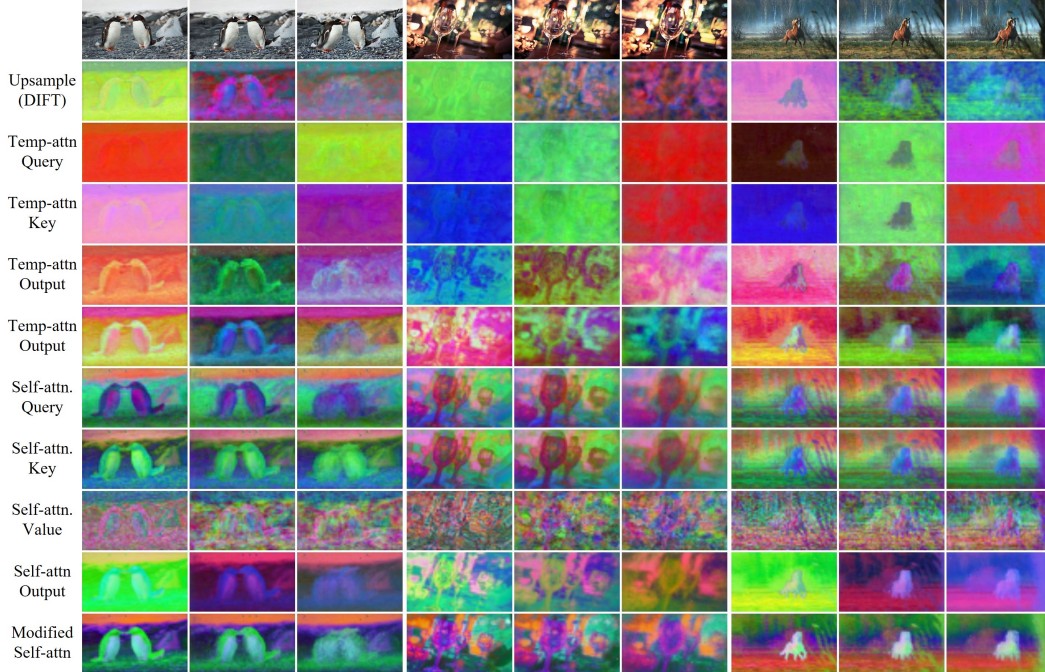

Figure 12: **Semantic correspondences of different features in video diffusion models.** We find features from self-attention layers to be more semantically aligned than that of temporal attention layers and upsampling layers, while our modified self-attention layer produces the most aligned results due to its explicit formulation to attend to the first frame.

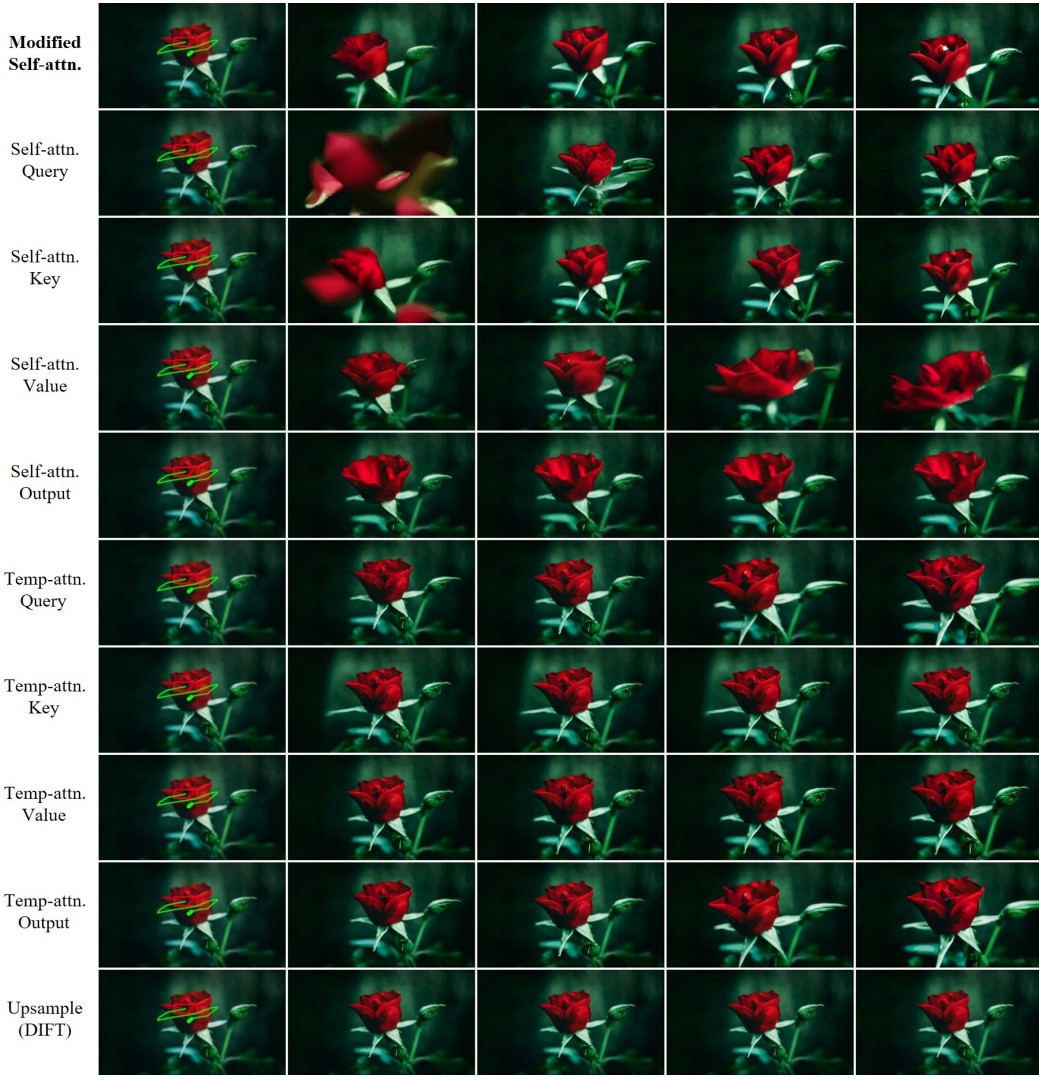

Figure 13: **Ablation on U-Net feature maps.** Applying loss on feature maps extracted from original self/temporal-attention layers or upsampling blocks fails to follow the trajectory due to the semantic misalignment across frames. In contrast, performing optimization with our modified self-attention layers can produce videos consistent with the input trajectory, indicating the importance of using semantically aligned feature maps. Please see *our project page* for more qualitative results.

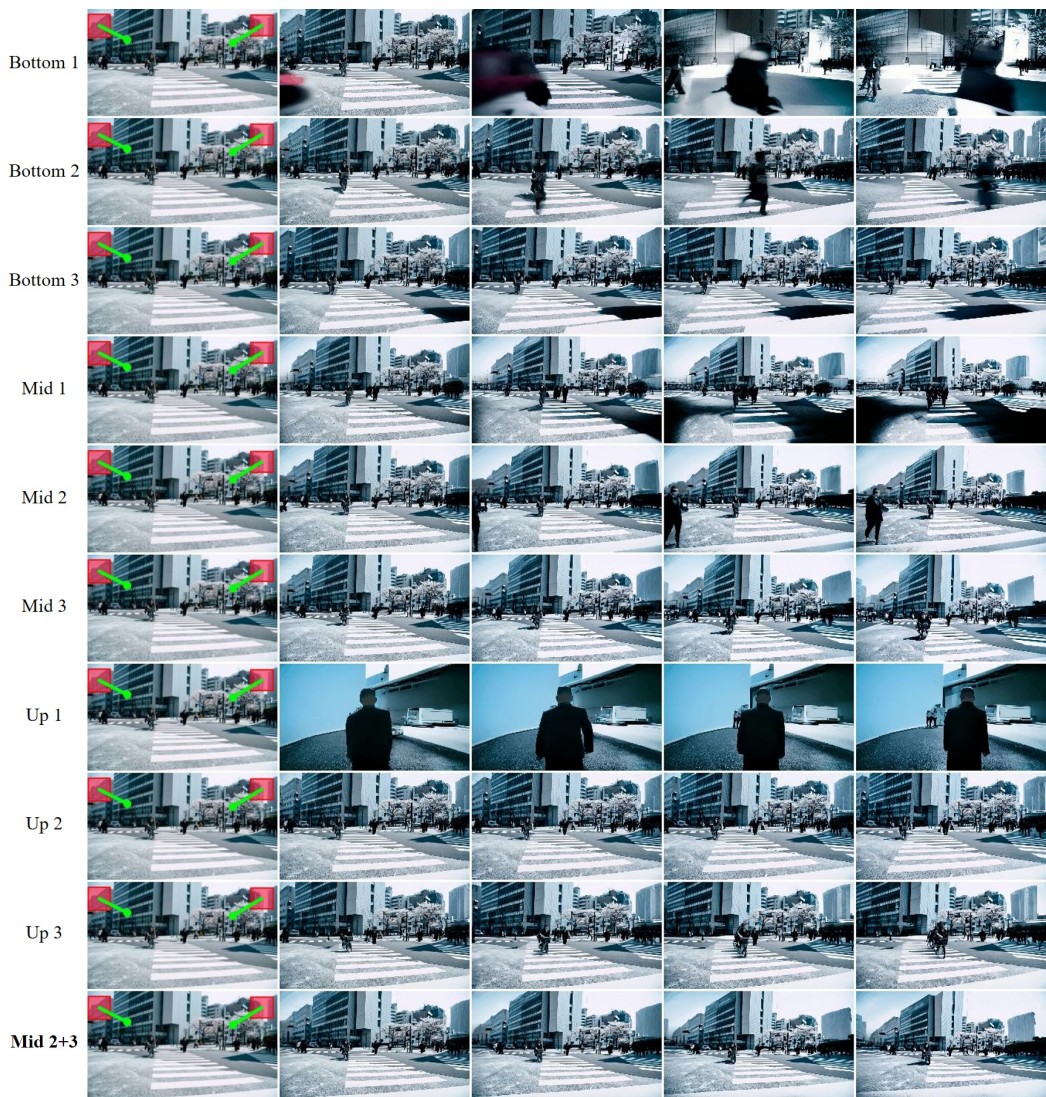

Figure 14: **Ablation on U-Net layer to extract feature maps.** Consistent with the quantitative results in Fig. 6, feature maps extracted from the middle resolution level are most useful for trajectory guidance. Optimizing on other feature maps may generate unrealistic videos with low motion fidelity.

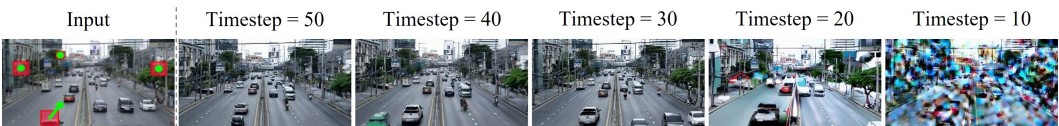

Figure 15: **Visual comparison of different denoising timesteps.** Here we show the *last* frame of the generated video. Optimizing latent at later denoising process leads to severe artifacts.

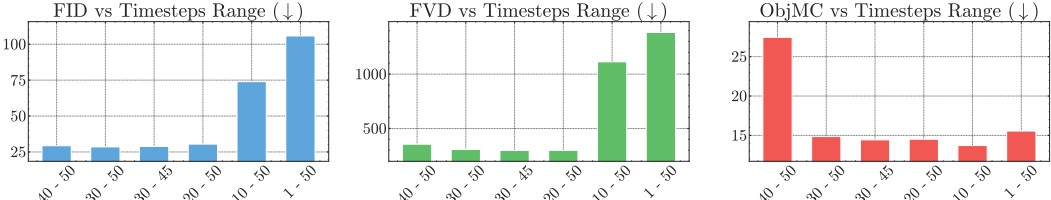

Figure 16: **Effect of optimizing latent at multiple denoising timesteps.** Here we perform optimization on multiple denoising timesteps ($t = 50$ corresponds to standard Gaussian noise). Similar to performing individual timestep Fig. 10, performing optimization up to middle timesteps (e.g. $50 - 30$) achieves the best motion fidelity while maintaining visual quality.

