# OpenReview forum: "SG-I2V: Self-Guided Trajectory Control in Image-to-Video Generation"
_ICLR.cc/2025/Conference — ICLR 2025 Poster_

### Official Review · Reviewer_JD8W · 2024-10-22

**Soundness:** 3
**Presentation:** 3
**Contribution:** 2
**Rating:** 6
**Confidence:** 4

**Summary:**

The paper introduces a novel framework for controllable image-to-video generation. The core innovation of SG-I2V lies in its ability to achieve zero-shot control over object motion and camera movement within generated videos, leveraging a pre-trained video diffusion model. The method does not require task-specific fine-tuning or additional external knowledge. The authors propose an approach that analyzes feature maps from the upsampling blocks of video diffusion models to identify weak semantic alignments across frames. To address this, they introduce a modified self-attention mechanism that aligns features across frames, enabling precise control over element positions in the generated video. The framework optimizes the latent space to enforce similarity between aligned features within bounding box trajectories, and a post-processing step is applied to maintain the expected distribution of high-frequency noise expected by the diffusion model. The paper demonstrates that SG-I2V outperforms unsupervised baselines and is competitive with supervised models in terms of visual quality and motion fidelity.

**Strengths:**

1. The authors have been inspired by tuning-free text-to-video generation methods and have proposed a similarly tuning-free approach for image-to-video tasks. This eliminates the need for computationally expensive fine-tuning procedures, making the framework more accessible and efficient.
2. The paper provides a thorough analysis of semantic feature alignment within a pre-trained image-to-video diffusion model.
3. The authors have provided a project page that offers a more intuitive presentation of their work.

**Weaknesses:**

1. While the paper claims the ability to control an arbitrary number of objects, it lacks experimental evidence to support this claim. The examples provided for controlling object motion are limited to simple scenarios with only 1-2 objects, which may not demonstrate the method's capability across more complex scenarios with multiple objects.
2. The method's robustness is questionable due to the fine-tuning of parameters specifically for the VIPSeg dataset. The learning rate, for instance, is set to a specific value (0.21), raising concerns about whether these parameters would be universally effective across various image-to-video examples without further adjustments.
3. It is unclear whether the proposed method could be adapted to other base video generation models beyond the one used in the paper. The transferability of the method to different models is an important consideration for its broader adoption.
4. The current method appears to support only linear trajectories, which may not be sufficient for more complex motion control. There is a need to explore whether the method can handle other shapes and longer, more complex trajectories.
5. Lacking of reference or analyze the relationship with related work on trajectory-based image or video generation, including TraDiffusion[1] and DragEntity[2].

[1] TraDiffusion: Trajectory-Based Training-Free Image Generation.

[2] DragEntity: Trajectory Guided Video Generation using Entity and Positional Relationships.

**Questions:**

See Weaknesses.

---

> ### Author Response · Authors · 2024-11-23
> **Official Comment by Authors**
>
> We thank the reviewer for the insightful feedback. In response, we have revised the paper and supplementary website, incorporating the reviewer’s suggestions. All modifications in the revision based on the reviewer’s feedback are highlighted in orange. In the following, we address each of the comments and questions.
>
> **W1 & W4: Complex nonlinear & longer trajectories with multiple objects.**
>
> We appreciate your constructive feedback. We have updated the website (https://sgi2v-paper.github.io/gallery_VIPSeg.html) to include results with non-linear trajectories with multiple (>2) constraints. We are working to add more examples and hope these results address your concerns.
>
> **W2: Parameter robustness.**
>
> Thank you for your insightful feedback. While we conduct hyper-parameters analysis on VIP-Seg, we directly apply the same set of hyper-parameters to generate qualitative results using web images. This demonstrates the generalizability of the hyper-parameters. One of the strengths of our approach is that hyper parameters can be easily adjusted without an expensive re-training procedure.
>
> **W3: Experiments on other image-to-video diffusion models.**
>
> Thank you for your suggestion. We are currently extending our experiments to include other types of video diffusion models, and hope to demonstrate this in future work. We kindly note the following:
>
> - SVD was chosen due to its wide application in previous image-to-video generation papers. Importantly, state-of-the-art open-sourced trajectory-controlled methods (at the time of submission) such as DragAnything and DragNUWA were all built on SVD.
> - SVD is unique in that it requires only images as inputs. Other image-to-video diffusion models usually require additional text prompts as inputs, making evaluation challenging because text prompts strongly influence motions.
>
> Given these considerations, we believe SVD is the natural choice for our experiments. We are exploring extensions of the method to emerging architectures, such as video diffusion transformers and hope to tackle this problem in future work.
>
> **W5: TraDiffusion and DragEntity**
>
> Thanks for bringing up these works. We have revised Section 2 to discuss TraDiffusion and DragEntity (highlighted in orange).
>
> Briefly:
>
> - TraDiffusion is a method for controlling spatial layouts in text-to-image generation. Its underlying techniques are similar to Self-Guidance[1], which is already discussed in the paper.
> - DragEntity is a supervised method for trajectory-controlled image-to-video generation. Its techniques are similar to DragAnything, which is discussed in the paper. As the authors did not make the code publicly available, we could not perform experiments with it.
>
> We hope these address your concerns, and we are happy to discuss further if needed!
>
> References:
>
> [1] Dave Epstein, Allan Jabri, Ben Poole, Alexei A. Efros, and Aleksander Holynskio. Diffusion Self-Guidance for Controllable Image Generation, 2023.

---

> > ### Comment · Reviewer_JD8W · 2024-11-24
> >
> > I appreciate the author’s response to my questions. I believe their work holds significant potential value for the community and is likely to be well-received. In my opinion, it could be considered for acceptance.

---

### Official Review · Reviewer_F4s1 · 2024-10-27

**Soundness:** 3
**Presentation:** 3
**Contribution:** 3
**Rating:** 6
**Confidence:** 5

**Summary:**

In this paper, the authors first analyze the semantic correspondences among frames in different latent. Then based on the observation, they propose SG-I2V, a training-free framework for trajectory control through aligning the self-attention with given bounding boxes based on pre-trained I2V model SVD. Additionally, SG-I2V supports arbitrary input images and any number of objects or regions of a scene.

**Strengths:**

1. Authors use PCA to visualize feature maps of SVD and demonstrate inspiring motivation of method design.
2. This method is training-free and thus does not require additional data to tune the modules. Through optimizing the latent, SG-I2V succeeds in generating trajectory-controllable videos with high fidelity.
3. Inspired by previous work, they apply a high-frequency resampling post-processing to preserve the visual quality. The ablation study demonstrates the effectiveness of this module.
4. Exhaustive ablation experiments demonstrate the rationality of the designed modules and parameters.

**Weaknesses:**

1. MotionCtrl is a well-known method for trajectory control and also provides the official code and pre-trained model for SVD (same backbone). SG-I2V is supposed to compare with MotionCtrl.
2. Since each inference requires individual optimization calculations, authors are supposed to discuss it. If the additional time is too much, is there any way to reduce it?

**Questions:**

Has the author tried this method on other I2V frameworks? Specifically, what is the generalizability of this method? Can it be adapted to other UNet models, such as DynamiCrafter? What about the DiT architecture?

---

> ### Author Response · Authors · 2024-11-23
> **Official Comment by Authors**
>
> We thank the reviewer for their insightful and positive feedback. In response, we have revised the paper. Below, we address each of the weaknesses and questions in your review.
>
> **W1. MotionCtrl with SVD backbone.**
>
> We appreciate the suggestion. After thoroughly reviewing the official MotionCtrl codebase, we could only find a camera-controlled model using the SVD backbone. Nevertheless, we have compared our method against other representative trajectory control models with SVD backbones that are designed specifically for dragging-based input, such as DragAnything and DragNUWA. We believe these comparisons sufficiently demonstrate the efficacy of our approach.
>
> **W2: Inference time.**
>
> Thank you for your question. Please refer to our general response for the result.
>
> **Q1: Experiments on other image-to-video diffusion models.**
>
> Thank you for your suggestion. We are currently extending our experiments to include other types of video diffusion models, and hope to demonstrate this in future work. We kindly note the following:
> - At the time of development, open-sourced, high-quality DiT-based image-to-video diffusion models (e.g., CogVideoX-I2V) were not publicly available.
> - SVD was chosen due to its wide application in previous image-to-video generation papers. Importantly, state-of-the-art open-sourced trajectory-controlled methods (at the time of submission) such as DragAnything and DragNUWA were all built on SVD.
> - SVD is unique in that it requires only images as inputs. Other image-to-video diffusion models usually require additional text prompts as inputs, making evaluation challenging because text prompts strongly influence motions.
>
> Given these considerations, we believe SVD is the natural choice for our experiments. We are exploring extensions of the method to emerging architectures, such as video diffusion transformers and hope to tackle this problem in future work.
>
> We hope these address your concerns, and we are happy to discuss further if needed!

---

> > ### Comment · Reviewer_F4s1 · 2024-11-23
> >
> > Hi,
> >
> > Thanks for your response. Although this method takes much more time for inference, I think we should pay more attention to the contributions of this paper. I vote for acceptance and I have given a fair score.

---

### Official Review · Reviewer_g4dj · 2024-10-29

**Soundness:** 3
**Presentation:** 3
**Contribution:** 2
**Rating:** 5
**Confidence:** 5

**Summary:**

The paper presents a zero-shot image-to-video diffusion framework for controllable motion generation. Specifically, the proposed SG-I2V exploits the self-guidance in the cross-frame attention feature maps to regulate object trajectory. To further improve the video quality, a post-processing with high-frequency noise injection is designed to ensure that the latent distribution is not disrupted in online latent optimization. Experiments conducted on the VIPSeg dataset verify the proposed approach on both object motion control and camera control.

**Strengths:**

1.	The proposed approach employs the feature map correspondences to regulate video latent updating and achieves object trajectory control in a zero-shot manner.
2.	Both of the quantitative and qualitative experimental results demonstrate the effectiveness of the proposed approach.
3.	The paper is well-written and the technical figures are very clear.
4.	The investigation and ablation studies (including the visualization of feature maps) on the feature map selection is comprehensive for proposal validation.

**Weaknesses:**

1.	The major concern for this paper is about the technical contribution. Even though the weakness of the unsatisfactory feature alignment in the up-sampling blocks (the generalization problem in video domain) has been revealed, the latent optimization is still borrowed from the drag-based control methods, e.g., DragDiffusion. The feature map selection and validation are valuable in my personal opinion. Nevertheless, there is no concept new in the whole architecture design. Therefore, it can be treated as another DragDiffusion of the video diffusion version. Moreover, the high-frequency noise initialization is also proposed in FreeTraj, which should not the key technical contribution in this paper.
2.	The motivation of involving Gaussian heatmap $\textbf{G}_{b}$ in the Eq. (1) is not clear.
3.	It is interesting that the performance trend of ObjMC is flickering as shown in the Figure 10. The reason behind this could be discussed and analyzed.
4.	Is there any additional computational (time and memory) cost for the latent optimization? Including some analysis of them will make the experimental section stronger.
5.	Are there any visualized comparisons on the VIPSeg dataset? All the presented cases seem be derived from the web images with high quality.

**Questions:**

Please see the weaknesses.

---

> ### Author Response · Authors · 2024-11-23
> **Official Comment by Authors**
>
> We thank the reviewer for the insightful feedback. In response, we have revised the paper and supplementary website, incorporating the reviewer’s suggestions. All modifications in the revision based on the reviewer’s feedback are highlighted in purple. In the following, we address each of the comments and questions.
>
> **W1: Novelty.**
>
> Please refer to our general response for a detailed discussion. We believe our paper offers several novel insights that are not obvious from previous works like DragDiffusion and FreeTraj. These contributions have the potential to significantly impact both the zero-shot and video-generation communities.
>
> **W2: Motivation of Gaussian heatmap.**
>
> We appreciate the reviewer’s feedback and revised the paper to clarify the motivation for using the Gaussian heatmap in Section 3.3, Appendix C, and Table 2 (highlighted in purple). Since we use a bounding box to identify the region to move, it may include background pixels not intended to be moved. The Gaussian weighting alleviates this issue by focusing on pixels at the center while reducing focus near the edges of the bounding box.
> As shown in Table 2, the use of Gaussian weighting slightly but consistently improves the performance in all metrics on VIPSeg. In addition, it only adds negligible overhead for memory and inference speed. Therefore, we adopt it in our method.
>
> **W3: ObjMC is flickering in Figure 10.**
>
> Thank you for pointing this out. We have added explanations to Section 4.4 (highlighted in purple) and the website (https://sgi2v-paper.github.io/gallery_timestep2.html). We note that ObjMC only measures trajectory accuracy without considering visual quality. As demonstrated on the website, when optimizing latents in early timesteps (i.e., t=[10, 1]) the generated video contains severe artifacts along the target trajectory (as indicated by the bad FID and FVD scores). Yet, CoTracker is robust enough to also track these artifacts along the trajectory, leading to a decreased ObjMC score.
>
> **W4. Inference time and memory.**
>
> Thank you for your feedback. Please refer to our general response for the result.
>
> **W5: Visualized results on the VIPSeg dataset.**
>
> Thank you for the great suggestions. Results on the VIPSeg dataset have been added to our website (https://sgi2v-paper.github.io/gallery_VIPSeg.html) and Appendix B. These results demonstrate that our method effectively handles multiple bounding boxes with nonlinear trajectories.
>
> If you have any further concerns, we are happy to discuss them.

---

> > ### Comment · Reviewer_g4dj · 2024-11-27
> >
> > Thanks for the authors' feedback and the detailed ablations. Most of my concerns have been addressed, e.g., runtime and visualization on VIPSeg. The remianed one is still about the technical novelty. I really appreciate the comprehensive investigations on the feature map selection, and I also admit that there are some technical distinctions, such as post-processing on high-frequency components, between this work and previous tuning-free controllable image/video generation. Nevertheless, the key idea, i.e., feature injection and calibration via latent optimization, is leveraged from the exisitng work DragDiffusion.
> >
> > Even though, I still think the abundent ablations (Figure 5, 6, 8, 9, 10) can give some insights for future research about the tuning-free feature guidance in diffusion. That could be more valueable in my opinion.
> >
> > I will consider change my rating and pay more attention on the core contribution.

---

> > > ### Author Response · Authors · 2024-11-29
> > > **Official Comment by Authors**
> > >
> > > We sincerely thank the reviewer for their thoughtful feedback. As acknowledged by the reviewer, our proposed method is not a simple adoption of DragDiffusion (a technique for image diffusion models) to the video diffusion models. Instead, our method houses several technically distinct and nontrivial contributions—such as modified self-attention mechanisms and high-frequency preserved post-processing—specifically tailored for video diffusion models. The effectiveness of these techniques has been rigorously validated through ablation studies, and we believe our findings have the potential to inspire future works in the zero-shot and video generation communities.
> > >
> > > Given these considerations, we would appreciate it if the reviewer could reconsider the rating.

---

> > > > ### Author Response · Authors · 2024-12-02
> > > > **Official Comment by Authors**
> > > >
> > > > We sincerely appreciate the reviewer’s diligent efforts in reviewing our work. Since the final deadline for receiving the reviewer's response is in 16 hours (December 2nd at 11:59 AoE), we would be happy to hear any remaining concerns that keep you maintaining the current rating. If the reviewer found our previous response satisfactory, we kindly ask you to consider raising your rating by the deadline.

---

### Official Review · Reviewer_AD9e · 2024-10-31

**Soundness:** 3
**Presentation:** 3
**Contribution:** 3
**Rating:** 6
**Confidence:** 4

**Summary:**

This paper proposes a training-free, self-guided trajectory control framework for image-to-video generation, named SG-I2V. The authors also design operations for semantically aligned feature map extraction, trajectory enforcement, and high-frequency preservation in post-processing, which together constitute the SG-I2V framework. Additionally, comprehensive experiments are conducted to demonstrate the effectiveness of SG-I2V and the impact of specific parameters in SG-I2V.

**Strengths:**

This paper exhibits several strengths:
1.The motivation behind the research is clear and well-founded.
2.The use of latent optimization for trajectory control appears both intuitive and innovative.
3.The experimental results validate the effectiveness of the proposed method on the SVD model. The findings are convincing, and the ablation study is thorough.

**Weaknesses:**

This paper exhibits several Weaknesses:
1.There is a lack of generalization verification across various types of video diffusion models, such as VideoCrafter, EmuVideo (based on the U-Net structure), and CogVideoX-I2V (based on Diffusion Transformer).
2.The issue of inconsistent features across frames, as discussed in the introduction, may not be universally applicable. This problem might not be evident in some of the latest 3D full attention-based video diffusion models (e.g., OpenSoraPlan 1.2, EasyAnimate V4, and CogVideoX) or when using VideoVAE.
3.There are errors in the formula expressions:
1)The symbols in Formula 1 are confusing. Does Fn represent an operation or a feature map? How does Bb,n (denoting [h, w, x, y]) perform matrix operations with feature maps (shape of [h, w, d])? What does “[Bb,n]” signify?
2)There are two instances of zt* in formula 2.
4.In line 194, the author emphasizes "Yet, all of these methods focus on text-based generation." However, MOFT[1] is also training-free and can achieve image-to-video motion control. In addition, MOFT incorporates the concept of optimizing denoising latents. Can the authors elaborate on the similarities and differences compared to MOFT in terms of functionality and methodology?

[1] Video Diffusion Models are Training-free Motion Interpreter and Controller.

**Questions:**

1.How long does it take for SG-I2V to infer a single example?

---

> ### Author Response · Authors · 2024-11-23
> **Official Comment by Authors**
>
> We thank the reviewer for their insightful feedback. We have revised the paper to address the reviewer’s comments. All modifications in the revision are highlighted in blue. In the following, we address each of the comments and questions in the review.
>
> **W1: experiments on other image-to-video diffusion models.**
>
> Thank you for your constructive suggestion. We are currently extending our experiments to include other types of video diffusion models, and hope to demonstrate this in future work. We kindly note the following:
> - At the time of development, open-sourced, high-quality DiT-based image-to-video diffusion models (e.g., CogVideoX-I2V) were not publicly available.
> - SVD was chosen due to its wide application in previous image-to-video generation papers. Importantly, state-of-the-art open-sourced trajectory-controlled methods (at the time of submission) such as DragAnything and DragNUWA were all built on SVD.
> - SVD is unique in that it requires only images as inputs. Other image-to-video diffusion models usually require additional text prompts as inputs, making evaluation challenging because text prompts strongly influence motions.
>
> Given these considerations, we believe SVD is the natural choice for our experiments. We are exploring extensions of the method to emerging architectures, such as video diffusion transformers and hope to tackle this problem in future work.
>
> **W2: 3D full attention-based video diffusion models**
>
> We are actively investigating application of our method to emerging video diffusion models and hope to tackle this in future work. Please kindly note that, at the time of development, high-quality 3D attention-based video DiT models were not publicly available.
>
>
> **W3: Error in formula**
>
> Thank you for identifying this issue. We have corrected the error in Equation 2 and updated Section 3.3 to improve the clarity of each term in Equation 1 based on your feedback.
>
> Specifically:
>
> - $\tilde{F}_n(z_t) \in \mathbb{R}^{h \times w \times d}$ represents feature maps extracted from the modified self-attention layer, given the noisy latent $z_t$ as input to the model.
>
> - $\\tilde{F}\_n(z\_t)[\\mathcal{B}\_{b, n}] \\in \\mathbb{R}^{h\_b \\times w\_b \\times d}$ denotes feature maps cropped by the bounding box $\mathcal{B}_{b, n}$.
>
> These updates are highlighted in blue in the revised manuscript.
>
> **W4: Comparison with MOFT**
>
> We have included results and discussions regarding MOFT in Section 4.1, Table 1, and Appendix A (highlighted in blue). Key differences between our method and MOFT are as follows:
>
> - MOFT primarily focuses on reference-based motion control, requiring videos with target motions as references. In contrast, our method does not rely on reference videos, offering more flexibility and precise controllability.
> - MOFT optimizes on their proposed feature maps, claimed to be motion consistent (i.e., pixels belonging to the same motion exhibit similar vectors). However, semantic consistency is not guaranteed and it is not clear whether the same objects having different motions across frames have consistent feature vectors. As demonstrated in Table 1, we find optimizing on their proposed feature maps is less effective compared to our approach.
>
> **Q1. Inference time.**
>
> Thank you for your question. Please refer to our general response for the result.
>
> We hope these address your concerns, and we are happy to discuss further if needed!

---

> > ### Comment · Reviewer_AD9e · 2024-11-27
> >
> > Thanks for the response. The rebuttal has resolved my most concerns.

---

### Official Review · Reviewer_StsS · 2024-11-01

**Soundness:** 2
**Presentation:** 2
**Contribution:** 2
**Rating:** 5
**Confidence:** 5

**Summary:**

The paper introduces SG-I2V, a self-guided method designed to achieve zero-shot controllable image-to-video (I2V) generation. Technically, SG-I2V first extracts feature maps from the self-attention module during the video denoising process. To ensure that the generated video’s motion aligns with the given trajectory, the input latent code is optimized by enforcing cross-frame feature consistency along the bounding box trajectory. Furthermore, to preserve the quality of the generated video, the authors propose a post-processing method that retains the high-frequency components of the original latent code. The evaluation on the VIPSeg dataset demonstrates the effectiveness of the proposed method.

**Strengths:**

S1: The paper analyzes the intrinsic relationship between the motion in the generated video and the UNet feature maps.

S2: Building on this analysis, a self-supervised framework is designed to achieve zero-shot controllable image-to-video (I2V) generation.

**Weaknesses:**

W1: The performance values reported in Table 1 on the VIPSeg dataset differ from those in the DragAnything paper, which reports ObjMC, FVD, and FID values of 305.7, 494.8, and 33.5, respectively. Additionally, Table 1 shows DragNUWA outperforming DragAnything, which contradicts the findings in the DragAnything paper. Could the authors clarify the specific versions of DragAnything and DragNUWA used in their experiments, any differences in their evaluation setup compared to the original papers, and whether they used the same evaluation code and metrics? This information would help pinpoint the source of these discrepancies and ensure a fair comparison.

W2: Generally, in the field of video generation, the FVD and FID metrics often exhibit similar trends. I’m curious why SG-I2V performs worse than DragNUWA in FVD (by approximately 45 points) while showing better performance in FID (by about 2 points). Additionally, there seems to be a notable gap between SG-I2V and the best supervised method, with differences of around 45 in FVD and 3.6 in ObjMC. This disparity is quite significant, yet the paper describes the results as "competitive," which appears to be an overstatement.

W3: There is a discrepancy between Table 1 and Figure 4. According to the results in Table 1, SG-I2V performs worse than DragNUWA, while Figure 4 suggests that SG-I2V performs better, leading to contradictory information between the two experiments. Additionally, since DragAnything focuses on entity control, it may be more objective to include a comparison of entity control performance. Furthermore, the primary focus of the comparisons in this paper seems to be on zero-shot methods. Therefore, it would be beneficial for Figure 4 to include a comparison of visual quality with those zero-shot approaches as well.

W4: In the zero-shot baselines, the methods FreeTraj and DragDiffusion are not complete versions of the original models, therefore, more implementation details should be provided. Additionally, the original DragDiffusion model supports drag-based image editing, making it necessary to directly present the model’s performance results in video generation (by dividing the trajectory into 15 segments and running DragDiffusion on each segment individually).
Furthermore, since DragDiffusion requires an edit region as input, how should this input be provided appropriately to ensure a fair comparison?

W5: Some modules may not appear particularly novel from a technical perspective. For instance, the method of optimizing the input latent code to control the layout of video generation is a technique that has been widely utilized in various zero-shot controllable generation papers.

**Questions:**

Q1: The paper proposes optimizing the input latent code and applying high-frequency preserved post-processing between steps 30 and 45 during the denoising process. The inference time required to generate a 14-frame video should be reported.

Q2: I am curious why the modified spatial self-attention mentioned in Section 3.2 shows better alignment with the initial frame. A more detailed explanation would be helpful. Additionally, after modifying the self-attention in SVD, there is a gap between the modified inference pattern and the original learnt pattern, could this affect the quality of video generation? It would be beneficial to evaluate the change in FVD for I2V generation before and after modifying the self-attention in SVD.

Q3: I hope the authors can address the concerns raised in the "Weaknesses" section.

---

> ### Author Response · Authors · 2024-11-23
> **Official Comment by Authors 1/2**
>
> We thank the reviewer for their insightful feedback. We have revised the paper, website, and supplementary material, incorporating the reviewer’s suggestions. All modifications in the revision are highlighted in red. In the following, we address each of the comments and questions in the review.
>
> **W1: Performance gap with DragAnything.**
>
> We thank the reviewer for pointing this out. We corresponded with the authors of DragAnything and verified that there is no error in our evaluation; the metrics differ due to implementation details, such as how image resolution is handled. As the DragAnything authors did not release their complete evaluation pipeline, we could not perfectly reproduce their implementation. We describe the differences as follows.
>
> For **ObjMC**,
>
> - **Resolution of video:** Our evaluation uses a uniform video resolution of $320\times 576$, while DragAnything resizes the generated videos to the original resolution, which varies between $480×800$ and $1440×256$. Since ObjMC computes error distance, the resolution difference affects the scale of the reported value.
> - **Invalid Ground Truth Trajectory points:** Some ground truth trajectory points in the dataset contain invalid coordinates (e.g., objects moving out of frame). As all the baselines, including ours, can not be conditioned on these points, we exclude them from evaluation.
>
> For **FID and FVD**,
>
> - DragAnything uses 25 inference steps in their paper. To ensure a fair comparison with baselines, we use longer steps (50) when running their code. This contributes to their improved FID and FVD in our table.
>
> To ensure reproducibility, we have added specific evaluation details and model versions (if applicable) in Table 1 and Appendix A (marked in red). Furthermore, we have uploaded our evaluation scripts in the supplementary material and will publicly release the evaluation scripts upon acceptance.
>
> **W2: Performance gap in FVD and ObjMC with supervised baselines.**
>
> FID focuses on per-frame visual quality, while FVD is more sensitive to the motion smoothness of the video. As a tuning-free method, we generate videos at the original resolution of SVD (576x1024) and then downsample them to match the output resolution of the baselines for FVD computation. In contrast, the supervised baselines fine-tune SVD to generate lower-resolution videos directly (320x576). Generating smooth motion at a higher resolution is generally harder, and this discrepancy in evaluation settings contributes to the different trends between FID and FVD.
>
> We also toned down claims of “competitive performance” with supervised baselines in the abstract, introduction, and Table 1.
>
> **W3: Discrepancy between Table 1 and Figure 4.**
>
> The results in Table1 show a different trend than the qualitative results in Figure 4 because Figure 4 aims to highlight the limitations of supervised baselines and show how our tuning-free method generalizes to these failure cases. For instance, DragAnything is fine-tuned on entity-level trajectories and struggles with part-level control, whereas our tuning-free method can cover these cases by benefiting from the generalizability of the underlying image-to-video diffusion model. To avoid confusion, we have clarified the purpose of Figure 4 in its caption and Section 4.3 (marked in red). The general qualitative results can be found on our supplementary website: https://sgi2v-paper.github.io/gallery_baseline.html and https://sgi2v-paper.github.io/gallery_baseline2.html.
>
> Additionally, we have added qualitative comparisons with zero-shot baselines in Appendix B and on our supplementary website https://sgi2v-paper.github.io/gallery_zero.html. These results highlight our advantage over zero-shot baselines
>
> **W4: Implementation details of zero-shot baselines.**
>
> We appreciate the reviewer’s suggestions and have added implementation details for the DragDiffusion and Freetraj baselines in Appendix A (marked in red). We are also conducting additional experiments with the original DragDiffusion, which we hope to be presented during the rebuttal. For a fair comparison, we treat the edit region of DragDiffusion as the entire image space since our model does not take the edit region into account. We kindly note that the original DragDiffusion, which was designed for image editing, may produce temporally inconsistent frames (e.g., modifying scene contents rather than moving objects or the camera, as shown in an example on their webpage https://yujun-shi.github.io/projects/drag_diffusion_resources/oilpaint_mountain.mp4). This will limit performance in our video generation setting.

---

> > ### Author Response · Authors · 2024-11-23
> > **Official Comment by Authors 2/2**
> >
> > **W5: Novelty of latent optimization.**
> >
> > Thank you for your feedback. Please refer to our general response for a detailed discussion. Although latent optimization was previously used in text-to-image generation, adopting it to image-to-video generation presents unique challenges. We believe our paper offers several novel insights into extending latent optimization for image-to-video generation that are not obvious from previous latent optimization works like DragDiffusion. These contributions have the potential to impact both the zero-shot and video-generation communities.
> >
> > **Q1. Inference time.**
> >
> > Thank you for your question. Please refer to our general response for the result.
> >
> > **Q2. The role of modified self-attention.**
> >
> > The key issue with SVD is the decoupled design of spatial and temporal attention, which causes semantic inconsistencies in feature maps across frames. In our modified self-attention, each frame attends to the key and value tokens of the first frame. Thus, the attention output at each frame is a weighted combination of the value vectors of the first frame, which are more correlated.
> > Regarding its impact on the generation quality, we kindly note that this modification occurs during the loss computation and does not affect the denoising steps, preventing degradation in visual quality. We have modified Section 3.2 to clarify this point.
> >
> > If you have any further concerns, we are happy to discuss them.

---

> > > ### Comment · Reviewer_StsS · 2024-11-26
> > >
> > > Performance gap with DragAnything: The authors' evaluation method differs from previous methods in some details. While they emphasize that their evaluation approach is right, the significant discrepancy between their reported values and the original results remains a concern for me (e.g., the FVD for DragAnything in the original paper is 495, whereas in this paper, it is 268).
> > >
> > > Performance gap in FVD and ObjMC with supervised baselines: The authors attribute the differing FVD results to variations in video resolution, which they argue can affect the values. However, FVD is a critical metric for evaluating video motion, and videos are typically resized to 256x256 for FVD calculation. The substantial discrepancy in FVD results raises concerns about whether this method is comparable to supervised methods, either in terms of metric evaluation or visual quality.  Furthermore, the proposed method incurs significant computational overhead, with inference time extending to 5 minutes compared to approximately 30 seconds for the original DragAnything.
> > >
> > > I appreciate the authors' effort in providing various baseline comparisons. However, based on the presented results, the outcomes are not sufficiently convincing. Some results do not demonstrate improvements (e.g., cases involving the dog, hair, coffee, and the zoom-out shot of the burning book).
> > >
> > > In summary, I have carefully reviewed the comments from other reviewers as well as the authors' responses. Overall, while I acknowledge the authors' engineering efforts, the improvements achieved are not sufficiently compelling given the significant increase in inference cost (e.g., as highlighted in the metric and demo comparisons above). Therefore, I intend to maintain my score.

---

> > > > ### Author Response · Authors · 2024-11-27
> > > > **Official Comment by Authors**
> > > >
> > > > We thank the reviewer for their insightful response. In the following, we address the points raised in the response.
> > > >
> > > > **Performance gap with original DragAnything.**
> > > >
> > > > Please kindly note that the complete evaluation pipeline of DragAnything is not publicly available; we have been in correspondence with the authors but unfortunately cannot reproduce the original reported results despite our best efforts. Regarding FVD, the original DragAnything runs 25 inference steps in their paper. Since our method uses 50 inference steps, we rerun DragAnything with longer steps (50), contributing to improved FID and FVD from the originally reported results. We validated that our results are consistent across both the PyTorch and TensorFlow implementations. **For full transparency, we have uploaded our evaluation codes in the supplementary material and will publicly release additional scripts for fully reproducing our quantitative results upon acceptance.** Furthermore, the trends observed in FVD, FID, and ObjMC metrics are consistent with our visual results (demonstrated on the website) in the ablation studies. Thus, we stand by all our quantitative and qualitative results.
> > > >
> > > > **FVD evaluation.**
> > > >
> > > > Thank you for highlighting this point. Indeed, we downsampled all the generated videos, including those from our method and the supervised baselines, to a resolution of 256×256 for FVD evaluation. For clarity, this explanation has been added to Appendix A. Additionally, the corresponding implementation can be found in eval_FVD.py, which is included in the supplementary material.
> > > >
> > > > **Inference time.**
> > > >
> > > > Our key contribution is significantly narrowing down the performance gap between zero-shot methods (e.g., our proposed approach) and supervised methods (e.g., DragAnything). While supervised methods offer faster inference time (e.g., 90 seconds), they require annotated datasets and long training time (e.g., several days). In contrast, zero-shot methods, while requiring longer inference time (e.g., 5 minutes), eliminate the need for annotated datasets and training (e.g., 0 minutes). Another benefit of zero-shot methods over the fine-tuned method is that we can generate videos at the original resolution ($576 \times1024$), while many supervised baselines (e.g., DragAnything) finetunes at a lower resolution ($320 \times 576$) due to the high computation cost. Thus, our work addresses a different point in this tradeoff space than supervised approaches.
> > > >
> > > > **Some results presented on the website do not demonstrate improvements over supervised baselines.**
> > > >
> > > > We kindly note that the objective of our project is to bridge the performance gap between supervised methods (e.g., DragAnything) and zero-shot methods (e.g., our proposed method). The baseline comparisons presented on the website indicate that our approach achieves comparable visual and motion quality to supervised methods without requiring fine-tuning.
> > > >
> > > > If you have any further concerns, we are happy to continue discussing them.

---

> > > > ### Comment · Reviewer_F4s1 · 2024-12-01
> > > > **Question of "FVD is a critical metric for evaluating video motion"**
> > > >
> > > > Sorry, but I hope you don't mind my joining this discussion. As another reviewer, I read your points and found them to be very detailed and accurate. However, I think FVD is a critical metric for evaluating video quality but not for video motion. FVD primarily measures the distribution distance between two datasets. In motion control tasks, the dynamics of generated videos often differ significantly from those in the reference videos. This discrepancy can lead to an interesting phenomenon: videos with less motion tend to achieve better FVD scores. A simple way to observe this is by comparing FVD results in motion control papers with those in base model papers. Typically, motion control tasks report higher FVD values due to the increased dynamics in their generated videos. Despite this limitation, FVD remains a classic and valuable indicator for evaluating video quality. My point is that we need it but we can't blindly pursue the best FVD.

---

> > > > > ### Comment · Reviewer_StsS · 2024-12-01
> > > > >
> > > > > Thank you for your kind participation in the discussion. Firstly, I agree with some of your points, such as the fact that the dynamics and reference video in generative tasks can indeed differ (for example, in text-to-video generation). However, I disagree with your statement: "videos with less motion tend to achieve better FVD scores." Since the task of this paper is motion-controllable image-to-video generation with trajectories, where the appearance is provided by the input image and the motion is specified by the input trajectory (extracted from the reference video), the generated video should ideally be as close as possible to the original video (a lower FVD indicates that both the motion and appearance of the generated video are closer to the original).
> > > > >
> > > > > Therefore, FVD is not a metric that can be ignored. Furthermore, this paper seems to perform not better than previous methods in terms of both FVD and video quality (as shown on the project page). While this paper proposes a training-free method, the core technology is not particularly novel and does not offer much new insight. Additionally, it introduces a new drawback by requiring 5 minutes of inference time. If the video quality (both visual quality and evaluation metrics) is not better than the previous baseline, there is little technical innovation, and it introduces a new issue of increased inference time consumption, why should this paper be considered for acceptance?

---

> ### Comment · Reviewer_F4s1 · 2024-12-01
>
> Thanks for your reply.
>
> 1. For FVD, assume we have four datasets, A, B, C, and D, with dynamic degrees 50%, 55%, 85%, and 90%. When their quality is similar, there is no doubt that A (50%) and B (55%) have the best FVD. In your assumption, C (85%) and D (90%) should have better FVD than B (55%) and D (90%). However, **in Table 1, SVD (No Control) achieves a competitive FVD rather than the worst FVD**, which is not aligned with your assumption. I also have observed some similar phenomena in my previous motion-control-related work. In summary, sometimes average results will get higher scores. Another example is that in super-resolution tasks, sometimes the results with details close to the ground truth have worse PSNR indicators than slightly blurred results. Sorry for the naive examples here. **I just wanted to share some of my observations about FVD and I never ask you to ignore FVD.**
>
> 2. The upper limit of tuning-free methods is definitely lower than that of training-based methods. However, training-based methods need more data, calculation resources, and most importantly, re-training for each base model. Nowadays, the iteration of video generation models is very fast, and the video quality generated by these methods depends to a large extent on the quality of the basic model. Tuning-free methods can quickly be adapted to a new base model. If authors can apply their results to CogVideoX, their results should be more competitive unless DragAnything trains a new version based on CogVideoX. Based on this consideration, I think the author's method is still competitive among tuning-free methods.
>
> 3. I never question your rating. All reviewers give scores around borderline and I don't think they are obviously diverging. The questions you raise are all very valuable, and addressing them will undoubtedly make the article better. If authors want to change your rating, they should put more effort into addressing your concerns.

---

> > ### Author Response · Authors · 2024-12-02
> > **Official Comment by Authors**
> >
> > We sincerely appreciate reviewers StsS and F4s1 for their thoughtful discussions regarding FVD and our paper's contributions. Below, we address the concerns raised by reviewer StsS:
> >
> >
> > > **Accuracy of FVD evaluation**
> >
> > We kindly remind the reviewer that the reported metrics for all the baselines and our method were computed using the **same evaluation protocols**. For full transparency, we have included the evaluation scripts in the supplementary material. Additionally, the trends observed in the FVD, FID, and ObjMC metrics are consistent with the visual results presented on our website in the ablation studies. We therefore think all the reported values are accurate.
> >
> >
> > >  **Visual quality**
> >
> > We encourage the reviewers to evaluate visual quality comprehensively by considering both quantitative metrics (FID, FVD) and qualitative results available on our website (https://sgi2v-paper.github.io/gallery_baseline.html, https://sgi2v-paper.github.io/gallery_baseline2.html). Many examples on our website demonstrate that our fine-tuning-free method performs better than or on par with supervised baselines in visual quality. Moreover, our method achieves superior FID scores compared to these baselines. Considering that FID and FVD are not perfect metrics for assessing visual quality, we believe it is reasonable to conclude that the difference in visual quality between our method and the supervised baselines is marginal.
> >
> >
> > > **Why should this paper be accepted?**
> >
> > As a zero-shot method that requires no fine-tuning, we believe our paper offers surprising and valuable insights to the ICLR community. Importantly, **these insights are non-trivial and cannot be easily predictable from prior work**:
> >
> > - We are the first to present high-quality results for zero-shot trajectory-controlled image-to-video generation. The relatively small performance gap with supervised baselines is remarkable, and our supplementary website demonstrates many examples where our approach outperforms the supervised baselines. **Achieving this level of quality without fine-tuning is unexpected and can inspire future work in this area**.
> >
> > - Achieving these high-quality results is challenging. **Our work introduces several technical insights that go beyond existing approaches**. Moreover, these techniques have the potential to benefit future research. For instance, the combination of high-frequency-preserving processing with latent optimization could be applied to other latent optimization methods to enhance visual quality.
> >
> > - As reviewer g4dj acknowledged, our **thorough analysis** of timesteps and feature maps is a valuable contribution to zero-shot and video-generation communities. For instance, our insights into the role of timesteps in motion generation could inspire the development of subsequent video generation work.
> >
> > - Both **zero-shot and fine-tuning-based approaches have distinct strengths and limitations**. As noted by reviewer F4s1, fine-tuning-based methods demand computationally expensive training on annotated datasets, which are not required by zero-shot approaches. We think these fundamentally different approaches should be assessed separately and encourage reviewers to focus on the value of the insights our paper offers, rather than solely comparing its performance to supervised baselines.
> >
> > As stated in the ICLR reviewer guideline  https://iclr.cc/Conferences/2025/ReviewerGuide, achieving state-of-the-art performance is not a mandatory criterion for acceptance. As a zero-shot approach that significantly reduces the performance gap with supervised baselines, we hope the reviewer will consider the value of the insights our paper brings to the zero-shot and video generation communities.
> >
> > Lastly, we appreciate reviewer F4s1’s suggestion to conduct experiments with newly emerged video diffusion transformers. Due to the rebuttal timeline constraints (please note that we are not allowed to present new experiment results after November 27th), we plan to conduct these experiments as future work. We would also like to remind reviewers that high-quality DiT-based image-to-video diffusion models (e.g., CogVideoX-I2V) **were not publicly available at the time of development**, and SVD was the natural choice for our experiments given all the state-of-the-art open-sourced trajectory-controlled baselines were built on SVD at the time of submission.

---

> > > ### Comment · Reviewer_StsS · 2024-12-03
> > >
> > > Thank you to the authors for their kind responses and to the other reviewers for the constructive discussions. I sincerely appreciate the authors' efforts during the rebuttal process. My views have already been shared previously, I think this paper is not good enough. Given this, I have voted a fair score and will leave the final decision to the area chair.

---

### Author Response · Authors · 2024-11-23
**General Response**

We thank the reviewers for their helpful feedback and insightful comments.

We are glad that the reviewers find our paper “well written” (g4dj), our motivation “well-founded,” and our method “intuitive and innovative” (AD9e). Reviewers StsS, g4dj, F4s1, and JD8W all highlighted our feature map analysis of SVD, which motivates our approach. Additionally, our experiments and ablation studies were recognized as “convincing”(AD9e), “thorough” (AD9e, JD8W), “comprehensive”(g4dj),  “valuable” (g4dj), and “exhaustive” (F4s1).

We have revised the paper and project page to incorporate the reviewers’ feedback. Here is the summary of the modifications we have made:
- We have included a discussion on inference time (StsS, AD9e,g4dj, F4s1)
- We have added qualitative results on more complex trajectories (https://sgi2v-paper.github.io/gallery_VIPSeg.html) (g4dj, JD8W)
- We have included a detailed explanation of our evaluation settings and uploaded the evaluation code for reproducibility (StsS)
- We have corrected errors in Equation 2 (AD9e) and improved the clarity of paper by rephrasing sentences across all the sections (AD9e, StsS, g4dj)
- We have added MOFT as an additional baseline (AD9e) and discussions on additional references (JD8W)
- We have added new qualitative results for the ablation study (https://sgi2v-paper.github.io/gallery_timestep2.html) (g4dj)

Below, we include a response to general questions raised by the reviewers. For other questions, please see our response to individual questions below each review.

**Novelty of SG-I2V** (StsS and g4dj)

Although latent optimization was previously used in text-to-image generation, adopting it to image-to-video generation presents unique challenges. Our paper offers novel insights into extending latent optimization for image-to-video generation that are not obvious from the previous work.

Specifically:
- As acknowledged by reviewers StsS, g4dj, F4s1, and JD8W, we conduct a new analysis of feature maps in a pre-trained image-to-video diffusion model and identify an important issue that these feature maps lack strong semantic alignment across frames. This observation is surprising because semantic alignment is a commonly known property in text-to-image diffusion models.
- With this insight, we introduce a novel solution, modified self-attention, where query tokens from each frame attend to the key and value tokens of the first frame. Our results show that modified self-attention produces semantically aligned feature maps, which are crucial for achieving latent optimization.
- Visual artifact is a common issue in latent optimization-based approaches. To mitigate it, we introduce a post-processing technique that preserves the high-frequency components of the original latent. While similar post-processing has been studied in the context of latent noise initialization,  it has never been combined with latent optimization before. We would like to note that this integration is recognized by reviewer F4s1 as a strength of our work.

We believe our paper offers several novel insights that are not trivial compared to previous work. These contributions have the potential to impact both the zero-shot and video-generation communities.

**Inference time and memory consumption of SG-I2V** (StsS, AD9e, g4dj, and F4s1)

We include it in Appendix C. On the VIPSeg dataset, our method requires an average of 305 seconds using an A6000 48GB GPU, while the original SVD takes about 90 seconds. The peak GPU memory usage is 30 GB. The total number of iterations in our method (75 iterations) is similar to previous latent-optimization-based image generation methods, such as DragDiffusion (80 iterations). We expect that inference times could be significantly improved in the future, given ongoing research in speeding up diffusion models.

---

### Meta-Review · Area_Chair_gYnj · 2024-12-21

**Metareview:**

This work receives mixed reviews. While reviewers initially raised concerns regarding performance discrepancies, limited evaluation, technical novelty, and computational cost, the authors' detailed response and clarification have largely addressed these concerns. The paper's contributions, particularly in analyzing and leveraging semantic feature alignment for controllable video generation, are considered to be novel and effective. Therefore, it is recommended to be accepted. However, the authors are encouraged to address the remaining concerns regarding performance discrepancies, generalizability, and computational cost in their future work.

**Additional Comments On Reviewer Discussion:**

All reviewers actively participated in the discussion with authors. The authors' detailed response and clarification have addressed most of the concerns about evaluation and computational cost. Most reviewers consider this work is well motivated by the feature map analysis and provide insights in proposing modified self-attention for semantic feature alignment as novel and effective solution.

---

### Decision · Program_Chairs · 2025-01-22

Accept (Poster)